# Assessment of the accuracy in UV index modelling using the UVIOS2 system during the UVC-III campaign

Ilias Fountoulakis[1], Kyriaki Papachristopoulou[2], Stelios Kazadzis[2], Gregor Hülsen[2], Julian Gröbner[2], Ioannis-Panagiotis Raptis [3], Dimitra Kouklaki[4,5], Akriti Masoom[2], Natalia Kouremeti[2], Charalampos Kontoes[4], Christos S. Zerefos[1,6,7,8]

[1]Research Centre of Atmospheric Physics and Climatology, Academy of Athens, 10680 Athens, Greece
[2]Physikalisch-Meteorologisches Observatorium Davos, World Radiation Center (PMOD/WRC), Davos Dorf, Switzerland
[3]Institute for Astronomy, Astrophysics, Space Applications and Remote Sensing, National Observatory of Athens, Greece
[4]Institute for Environmental Research and Sustainable Development, National Observatory of Athens, GR15236 Athens, Greece
[5]Laboratory of Climatology and Atmospheric Environment, Department of Geology and Geoenvironment, National and Kapodistrian University of Athens, 15784 Athens Greece
[6]Biomedical Research Foundation of the Academy of Athens, 11527 Athens, Greece
[7]Navarino Environmental Observatory (N.E.O.), 24001 Messenia, Greece
[8]Mariolopoulos-Kanaginis Foundation, 10675 Athens, Greece

*Correspondence to*: Ilias Fountoulakis (i.fountoulakis@academyofathens.gr)

**Abstract.** The third campaign for the calibration and intercomparison of solar UV radiometers (UVC III) took place at Davos, Switzerland in June - August 2022. More than 70 radiometers participated in the campaign and measured side-by-side with the portable reference spectroradiometer QASUME. The UVIOS2 system is a flexible UVI modelling tool that can be exploited for different applications depending on the inputs. Thus, different combinations of satellite, reanalysis, and/or ground-based inputs were used to test the UVIOS2 performance when it is used as a tool for UVI nowcasting or for climatological studies. While UVIOS2 provided quite accurate estimates of the average (for the period of the campaign) UVI levels, larger deviations were found for individual estimates. The average agreement between the UVI from the UVIOS2 and QASUME was better than 1% for all the different sets of inputs that were used for the study. The range of the variability was of the order of 40% for instantaneous measurements (15 min), mainly due to the model's inability to capture the instantaneous effects of cloudiness, especially under broken cloud conditions. Under clear-sky conditions the model was found to perform much better, with the differences between the model estimates and the QASUME measurements being smaller than 12% for 95% of the studied cases. Even at the pristine environment of Davos, single scattering albedo (SSA) was found to contribute significantly to the modelling uncertainties under cloudless conditions. For Aerosol Optical Depth (AOD) of the order of 0.2 – 0.4 at 550 nm, the role of the SSA was found to be comparable to the role of AOD in the modelling of the UVI.

# 1 Introduction

Exposure to solar ultraviolet (UV) radiation is vital for many living organisms including humans (e.g., Caldwell et al., 1998; Erickson III et al., 2015; Häder, 1991; Häder et al., 1998; Juzeniene et al., 2011; Lucas et al., 2019) but can be harmful when it exceeds certain limits (Diffey, 1991). Exposure of the human skin to UV radiation is the main mechanism that drives the formation of vitamin D, which, in turn, contributes to the strengthening of the immune system (e.g., Lucas et al., 2019; Webb et al., 2022). Moderate exposure to UV radiation has many more benefits for human health that are not related to the formation of vitamin D, such as the contribution to the maintenance of a good mental health and the curation of various skin diseases (Juzeniene and Moan, 2012). Nevertheless, overexposure to UV radiation is the main environmental risk factor for non-melanoma skin cancer, and among the main environmental risk factors for melanoma skin cancer and cataract (WHO, 1994). Determination of optimal sun exposure behaviors is not a simple task and, additionally to the surface solar UV radiation availability, it also depends on the physiology of each individual person (e.g., Armstrong and Cust, 2017; Hoffmann and Meffert, 2005; Lucas et al., 2019; McKenzie and Lucas, 2018; Webb et al., 2018; Webb and Engelsen, 2006).

A commonly used quantity for human health purposes is the UV index (UVI) (Schmalwieser et al., 2017a; Vanicek et al., 2000), which is a metric of the efficiency of UV radiation to cause erythema to the human skin. Generally, smaller exposure times and more precaution measures are recommended with increasing UVI. UVIs smaller than 2 are considered low, UVIs of 8 – 10 are considered very high, and UVIs exceeding 10 are considered extreme. In the 1980s and the 1990s, public awareness was caused due to the severe ozone depletion over high and mid latitudes which, if continued, would result in extreme UVI levels over densely populated regions of our planet (van Dijk et al., 2013; Newman and McKenzie, 2011). Although the adoption and the successful implementation of the Montreal Protocol prevented further depletion of stratospheric ozone and the consequent dangerous UV levels (McKenzie et al., 2019; Morgenstern et al., 2008), the future evolution of the levels of surface solar UV radiation is still uncertain, mainly due to the uncertainties in the impact of climate changes on surface solar UV radiation (Bernhard et al., 2023; Zerefos et al., 2023).

Since the 1980s, national and international networks for the monitoring of the UVI have been established to ensure accurate and timely information of the public (Blumthaler, 2018; Schmalwieser et al., 2017). Maintenance of a station that provides reliable UV measurements demands properly trained personnel to run the station and application of strict calibration and maintenance protocols. Furthermore, there are prerequisites for the installation of such stations (e.g., power supply, safety). Thus, it is impossible to achieve UVI monitoring with global coverage from the ground. Progress in satellite monitoring during the last decades allowed the retrieval of the UVI on a global scale. Currently, the UVI has been estimated with high spatial and temporal coverage using various techniques and various satellite products (e.g., see Table 1 in Zerefos et al., 2023). One of the most widely used climatological UVI datasets is provided by the Tropospheric Emission Monitoring Internet Service (TEMIS). TEMIS provides clear-sky UV doses since 1960 and all-sky UV doses since 2004, that have been calculated using measurements from various satellite sensors (https://www.temis.nl/uvradiation/UVarchive.php; Zempila et al., 2017). Widely used climatological datasets of the UVI with global coverage have been also retrieved using measurements from the Total

Ozone Mapping Spectrometer (TOMS) (Herman et al., 1999), the Ozone Monitoring Instrument (OMI) (Tanskanen et al., 2006), and the TROPOspheric Monitoring Instrument (TROPOMI) (Lindfors et al., 2018). As a result of the rapid progress in Earth observation monitoring, the aforementioned climatological satellite-based UV products have been proven to be reliable over wide regions of the planet (e.g., Lakkala et al., 2020; Zempila et al., 2016, 2017), although biases of the order of 10 – 20% have been reported over complex and polluted environments, while uncertainties can be even larger over highly reflective terrains at high latitudes (e.g., Lakkala et al., 2020). The accuracy of satellite-based estimates is limited due to the finite width of the satellite pixel (Kazadzis et al., 2009) and the weakness of satellite sensors to accurately probe the lower troposphere (Bais et al., 2019). In particular, assumptions are made in the satellite algorithms to describe the complex interactions between radiation, aerosols and clouds, which increase the uncertainty in the retrievals. Uncertainties in the assumed aerosol properties (Arola et al., 2021; Parisi et al., 2021), inaccurate distinction of the effect of highly reflecting terrains and cloudiness (Bernhard et al., 2015; Lakkala et al., 2020b), and uncertainties in the description of cloud cover, especially over high-altitude sites (Brogniez et al., 2016; Schenzinger et al., 2023) are among the main uncertainty sources.

Meteorological services provide UVI forecasting that is usually based on meteorological forecasting in conjunction with radiative transfer models (e.g., Feister et al., 2011; Long et al., 1996; Roshan et al., 2020). The Copernicus Atmospheric Monitoring Service (CAMS) - Atmosphere for example, provides five days clear-sky and all-sky UVI forecasts on a global scale based on the synergistic analyses of Earth-observation data, weather prediction and chemistry model forecasts, and radiative transfer modelling (Peuch et al., 2022; Schulz et al., 2022). UVI forecasts are commonly governed by the uncertainties in the forecasted meteorological parameters, mainly cloudiness (e.g., Schenzinger et al., 2023). Geostationary satellites provide continuous, nearly instant information for cloudiness over wide regions of the planet (Derrien and Le Gléau, 2005), which can be used to provide more accurate UVI estimates in nearly real time (Kosmopoulos et al., 2020) or UVI climatological products (e.g., Arola et al., 2002; Fragkos et al., 2024; Verdebout, 2000; Zempila et al., 2017).

Monitoring and/or forecasting of the UVI at mountainous sites is exceptionally challenging. Complex atmospheric conditions and complex terrains increase the uncertainties in the modelling of the UVI, while calibration and maintenance of sensors is not easy due to difficulties in access, power supply, and harsh weather conditions. Nevertheless, UVI increases with altitude and can reach extreme levels, which makes this information valuable for the inhabitants and the visitors of such locations. For example, extreme UVI of ~20 has been recorded in the Bolivian Andes (Pfeifer et al., 2006; Zaratti et al., 2003). Elevated UVI levels have been also recorded at high-altitude deserts in Argentina (Piacentini et al., 2003), while UVI frequently exceeding 15 has been measured at Tibet (Dahlback et al., 2007). UVIs frequently exceeding 11 have been also measured at European alpine stations (Casale et al., 2015) as well as at high altitude locations in Northwestern Argentina (Utrillas et al., 2016). Depending on atmospheric and terrain conditions, increases of the surface solar UV radiation levels with altitude can range from a few percent per km (Chubarova and Zhdanova, 2013; Pfeifer et al., 2006; Rieder et al., 2010; Schmucki and Philipona, 2002; Zaratti et al., 2003) to 10-20% (e.g., Chubarova et al., 2016; Sola et al., 2008), or even to more than 30%/km when surface albedo also increases with altitude (Bernhard et al., 2008; Pfeifer et al., 2006). During summer (if snow is absent), UVI increases with altitude mainly due to decreased Rayleigh scattering (Allaart et al., 2004; Blumthaler et al., 1994; Sola et al.,

2008). In general, the change in the levels of the solar UV irradiance with altitude depends on atmospheric composition and has a strong wavelength dependence which is introducing difficulties in the modelling of the UVI at mountainous sites (e.g, (Dvorkin and Steinberger, 1999; Krotkov et al., 1998)). At very high-altitude (or/and latitude) sites, ice and/or snow may persist even in late spring and summer resulting in extremely high UV exposure (e.g., Schmalwieser et al., 2017b; Siani et al., 2008; Utrillas et al., 2016).

The continuous operation of ground-based networks that provide highly accurate information is necessary, not only for the information to the public, but also for the validation and the improvement of satellite based UVI climatological and forecast/nowcast products (e.g., Fountoulakis et al., 2020b). In addition to the strict maintenance, operation, and calibration protocols that must be applied by the monitoring stations operators (e.g., Fountoulakis et al., 2020a; Garane et al., 2006; Gröbner et al., 2006; Lakkala et al., 2008), participation of the instruments to field campaigns further ensures the high quality and the homogeneity of the measured UVIs at different stations (Bais et al., 2001; Hülsen et al., 2020). The uncertainty in the UVI measured by the most accurate spectroradiometers that serve as world references can reach 2% (Gröbner and Sperfeld, 2005; Hülsen et al., 2016). Broadband filter radiometers that are commonly used in regional, national, or international networks for UVI monitoring are affected by larger uncertainties. In the context of the solar ultraviolet filter radiometer comparison campaigns (UVC, UVC-II, and most recently UVC-III) that were organized by the Physikalisch-Meteorologisches Observatorium Davos, World Radiation Center (PMOD/WRC) in 2006, 2017, and 2022 many broadband radiometers measured side-by-side with the world reference QASUME (e.g., Hülsen et al., 2020; Hülsen and Gröbner, 2007). Analyses of the measurements by the 75 instruments that participated in UVC-II resulted in the estimation of a calibration uncertainty of 6%. The overall uncertainty in the measurements was larger, due to other factors, mainly the imperfect angular response of the radiometers (Hülsen et al., 2020).

Furthermore, Davos is one of the few mountainous sites in the world where both, highly accurate UVI measurements, and measurements of the main factors that determine the levels of the UVI at the surface (and can be used as inputs for its modelling) are available, which allows us to assess the efficacy of a state-of-the-art UVI model to produce estimates and reconstruct UVI series under such conditions.

The first version of the UVIOS (UV-Index Operating System) nowcasting system has been already described in Kosmopoulos et al., (2021). The system has been upgraded recently in order to achieve faster and more accurate simulations. The new, improved UVIOS2 radiative transfer scheme can be used either as a tool for UVI nowcasting and forecasting or for climatological studies, depending on the inputs. In this paper, the UVI that has been simulated using the new UVIOS2 system with different inputs is described and validated against very accurate ground-based UVI measurements that were performed during the UVCIII campaign. The world reference QASUME that operated during the campaign provides measurements that are ideal for the validation of UVIOS2 due to their high accuracy, which allows the identification of the uncertainties in the modelling of UVI by UVIOS2. Highly accurate ancillary measurements that were available at the same period also allow the identification of the uncertainty sources in the UVI modelling. The main targets of the study can be summarized as follows:

- Describe the upgrades in UVIOS2 relative to the previous (UVIOS) system.

- Quantify the uncertainties, and the main uncertainty factors, in UVIOS2 simulations during the UVCIII campaign, when it is used as a tool for UVI nowcasting and climatological analysis.
- Evaluate and discuss in depth the uncertainty factors in the modelling of UVI at complex topography sites such as Davos.
- Discuss the uncertainty in forecasted UVI with respect to the uncertainty in the measurements of filter radiometers and discuss what are the prerequisites for improved UVI modelling.

It must be clarified that the study refers to a snow-free period at Davos, and thus the uncertainties related to the parameterization of surface albedo, which may be significant for higher altitude sites even in the summer, are not quantified or discussed here. The paper is organized as follows. A description of the used data and methods is provided in Section 2. The results of the
140 analysis are discussed in Section 3, and the main conclusions are summarized in Section 4.

## 2 Methodology

The UVIOS2 system is a flexible tool that can be exploited for different applications depending on the inputs. It can be used either as a nowcasting/forecasting tool, or to perform climatological studies. The accuracy of the simulated UVI depends on the compromise between the achievement of realistic computational times (i.e., the spatial and temporal extent of the
145 simulations) and the use of the most accurate model inputs. In the context of this work, we assessed the accuracy of UVIOS2 when it operates for real time applications (i.e., default setup that is used to simulate the real-time UVI over Europe) and when it is used for climatological studies (i.e., using ground-based measurements or reanalysis data as inputs) at the mountainous environment of Davos, Switzerland during the UVC III campaign (Hülsen and Gröbner, 2023). Assessment of the accuracy in UVIOS2 forecasts is out of the scope of the present study.

**2.1 The UVC-III campaign**

The third International Solar UV Radiometer Calibration Campaign (UVC-III) took place at Davos, Switzerland (Figure 1; 46.8°N, 9.83°E, 1610 m a.s.l.) from 13 June to 26 August 2022, and was organized by the PMOD/WRC as part of the WMO/GAW program (Hülsen and Gröbner, 2023). The QASUMEII data (see Sect. 2.2) were used as reference for the calibration of the broadband radiometers during the campaign. QASUME and QASUMEII were frequently calibrated during
the campaign using a portable calibration system with 250 W lamps. The two spectroradiometers remained stable within ±1% for the campaign period and their measurements differed by less than 3%. Seventy-five solar UV broadband filter radiometers were shipped to Davos and participated in the campaign. The UVI was derived from the measurements of the participating instruments using the calibration factors provided by the operators and the calibration factors that were calculated at Davos, and then the UVI from the radiometers was compared to the UVI measured by QASUMEII. All participating instruments were
also characterized for their angular and spectral response.

Ancillary measurements of many parameters that are valuable for the determination of the factors that result in discrepancies between the simulations of UVIOS2 and the measurements were performed during the whole period of the campaign. In particular:

- Aerosol optical properties were measured by a CIMEL radiometer (and many other radiometers that operate at the site) that is part of the AERONET network (Holben et al., 1998).
- Total Ozone Column (TOC) was measured by a Brewer spectroradiometer (Kerr, 2010; Kerr et al., 1985).
- Global and direct total solar irradiance by pyranometers and a pyrheliometer.
- Hemispherical sky images from sky cameras.
- Cloud cover in octas by a pyrgeometer (Dürr and Philopona, 2004)

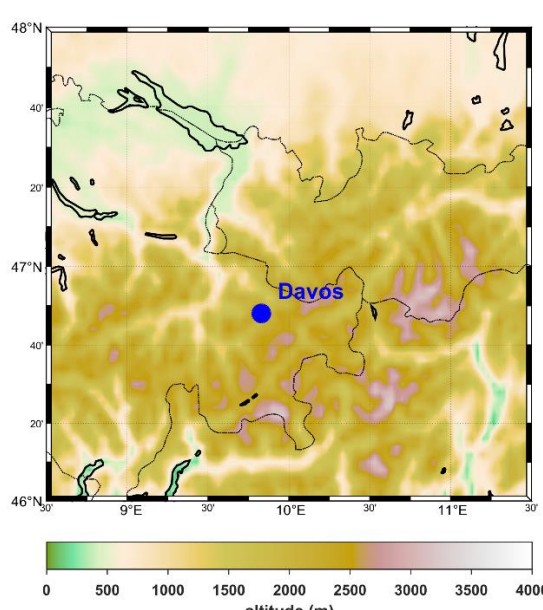

**Figure 1. Topographical map of Davos, Switzerland.**

## 2.2 QASUME

QASUME is a transportable spectroradiometer that is traceable to the scale of spectral irradiance established by the Physikalisch-Technische Bundesanstalt (PTB) and serves as a reference for spectral solar UV irradiance. The system is maintained by the PMOD/WRC and its measurement accuracy has been improved significantly in the last two decades. Since 2014, a second reference spectroradiometer (QASUMEII) is also operating and is used as an additional reference standard (Hülsen et al., 2016). Upgrades of technical characteristics and improved characterization methodologies have reduced the

expanded uncertainties in QASUMEII measurements at wavelengths above 310 nm from 4.8 % in 2005 (for QASUME) to 2.0 % in 2016 (Hülsen et al., 2016). More information about QASUME and QASUME II can be found in several relevant studies (Gröbner et al., 2005, 2006; Gröbner and Sperfeld, 2005; Hülsen et al., 2016). Both, QASUME and QASUMEII were measuring in the range 290 – 420 nm with a 15 min temporal resolution during the UVC-III campaign. These spectra were weighted with the erythema action spectrum (Webb et al., 2011) and were then integrated to calculate the erythemal doses, and subsequently the UVI (by dividing the dose rates in mW/m$^2$ with 25). For this work we have used only the UVI measured by QASUMEII, since the agreement between QASUMEII and QASUME is better than 3%.

## 2.3 The UVIOS2 system

UVIOS2 is built upon the UVIOS system (Kosmopoulos et al., 2021). The main change in the system configuration relative to the previous version is that the UVI is calculated in two steps:

    (i)        the UVI is calculated under cloudless skies and

    (ii)        the effect of clouds is quantified as a second step for the calculation of the all-skies UVI.

This change in the system's configuration was accompanied by two major modifications/upgrades: (1) the use of a more detailed UV look up table (LUT) for cloudless-sky calculations that increases the accuracy relative to the original version, and (2) the use of the UV cloud modification factor (CMFUV) concept used for the all skies UVI estimates. While the spectra for the LUT in the previous version of the model were simulated using the atlas plus modtran extraterrestrial spectrum (ETS), in the current version, the more recent QASUMEFTS (Gröbner et al., 2017) ETS was used. Furthermore, the ozone absorption cross sections by Bass and Paur (1980) were used to parameterize absorption by ozone, and not the Molina and Molina, (1986) that were used in the previous version. The variables that correspond to each of the five different dimensions of the LUT are listed in Table 1, along with their range and resolution. When SZA exceeds 89°, then UVI is considered equal to 0. When values of the other input parameters are above/below the limits shown in Table 1, then inputs are set to the upper/lower values of the used range. Such occasions are, however, very rare for mid-latitude sites.

**Table 1. Inputs of the LUT**

| Parameter | Range | Resolution |
|---|---|---|
| **Solar Zenith Angle (SZA) (°)** | 1 - 89 | 2 |
| **Total Column of Ozone (TOC) (DU)** | 200 – 600 | 10 |
| **Aerosol Optical Depth (AOD) at 550 nm** | 0 - 2 | 0.1 |

| | | |
|---|---|---|
| **Single Scattering Albedo (SSA)** | 0.6 - 1 | 0.1 |
| **Angstrom Exponent (AE)** | 0 – 2 | 0.4 |

The radiative transfer simulations for the creation of the LUT were performed using the UVSPEC model of the libRadtran version 2.0.4 package (Emde et al., 2016; Mayer and Kylling, 2005). Simulations were performed using the National Infrastructures for Research and Technology (GRNET) High Performance Computing Services and the computational resources of the ARIS GRNET infrastructure. Spectral simulations per 0.5 nm were performed for the spectral region 290 nm – 400 nm, using the QASUMEFTSETS (Gröbner et al., 2017) and the sdisort solver (Dahlback and Stamnes, 1991) which assumes pseudospherical atmosphere. Using a different ETS might result to differences in the simulated erythemal irradiances, as for example was shown in the study of Gröbner et al., (2017). Based on the results of the latter study we estimate that the simulated irradiances might differ by up to 5% if a different ETS was used, making the used ETS spectrum a major uncertainty factor in UVIOS2 cloudless simulations. Comparison with the UVIs that were simulated with LUT of the previous version of the model (i.e, where the atlas plus modtran ETS was used to construct the LUT) showed differences that were in all cases less than 2%. TOC is among the main regulators for the UVI levels at the surface and thus using TOC values that have been retrieved using different ozone absorption cross sections relative to those that have been used to create the LUTClick or tap here to enter text. would result in differences between the measured and the simulated UVI. Differences of 1 - 3% have been reported in the retrieved TOC depending on the used absorption cross sections (Fragkos et al., 2015; Redondas et al., 2014), which may result in differences of up to ~5% in the calculated UVI, depending mainly on the used cross sections, the SZA, aerosols load, and cloudiness (Blumthaler et al., 1995; Kim et al., 2013). In the domain for which the system is commonly used (i.e., Europe, North Africa, Middle East), variability in $SO_2$ and $NO_2$ has a minor impact on the UVI, and thus the total concentration of these species has been set to zero. The US standard atmosphere (Anderson et al., 1986) was used to describe the profiles of atmospheric state and composition, and the surface albedo was set to 0.05. Adjustment of the surface albedo to the local conditions when UVIOS is used over more reflective terrains (e.g., deserts, snow-covered surfaces) is within the model improvements that are planned for the future since under such conditions assuming a standard value of 0.05 could result in large uncertainties (e.g., Weihs et al., (2001)).

The optical properties profiles of libRadtran default aerosol model (Shettle, 1990) were scaled to the values of AOD (spectrally using the corresponding Ångström Exponent, AE) and SSA provided in Table 1. The AE and SSA values have been assumed to be invariant with wavelength for the simulations. Thus, we also did not consider the spectral dependence of the absorbing aerosol optical depth (as e.g., in the OMI and TROPOMI algorithms (Arola et al., 2021)), which may induce increased uncertainties over polluted regions (e.g., Roshan et al., (2020)). However, considering such information would increase significantly the size of the LUT and thus the computational time needed for the simulations, making the provision of the UVI in near-real time for wide areas impossible.

The UV spectra were weighted with the CIE (1998) action spectrum for the induction of erythema in the human skin (Webb et al., 2011) to calculate the UVI. Since all simulations have been performed for the average sea-surface level (i.e., altitude = 0 m, atmospheric pressure = 1013 mb). A correction for the effect of altitude, assuming an increase of 5% per km (e.g., Zempila et al., 2017) has been applied on the calculated UVI.

The cloud optical thickness (COT) from the Spinning Enhanced Visible and Infrared Imager (SEVIRI) instrument aboard the Meteosat Second Generation (MSG) satellites has been used to calculate the Cloud Modification Factor (CMF). The COT product is extracted operationally using the EUMETSAT Satellite Application Facilities of Nowcasting and Very Short-Range Forecasting, NWC SAF software package (Derrien and Le Gléau, 2005; Météo-France, 2016) and the broadcasted MSG data. A detailed description of the cloud products by MSG can be found in the relevant bibliography (Deneke et al., 2021; Météo-France, 2016). Using the MSG COT values and the SZA as inputs to the multiparametric equations described in Papachristopoulou et al. (2024) the shortwave CMF is calculated. Then, it is converted to CMFUV as described in Staiger et al. (2008). Finally, the UVI is calculated by multiplying the cloudless-sky values with the CMFUV. Using wavelength dependent CMFUV to simulate UV would be more accurate (Krotkov et al., 2001), but would increase computational time, and still the dominant uncertainty factor related with cloudiness would be the visibility of the solar disc.

To evaluate the methodology used for the quantification of the attenuation of the UVI by clouds the all-sky UVI was compared to QASUMEII measurements. Furthermore, the all-sky UVIs were compared to the corresponding values that were directly simulated by using cloud optical properties as inputs in the UVSPEC model of libRadtran. It was assumed that all low-altitude clouds over Davos extend from 4 km to 5 km (with reference to the a.s.l.), and all high-altitude clouds extend from 7 km to 8 km. High-altitude clouds were in all cases assumed to consist of ice crystals with effective radius equal to 20 μm and ice water content (IWC) of 0.005 g cm$^{-3}$, while low-altitude clouds were assumed to consist of water droplets with effective radius equal to 10 μm and liquid water content (LWC) value of 1 g cm$^{-3}$. The COT at 550 nm product from MSG was used as an additional input, which leads to an adjustment of the default LWC and IWC values, using the parameterizations by Hu and Stamnes (1993) for water and by Fu (Fu, 1996; Fu et al., 1998) for ice clouds. The latter simulations were performed for the altitude of the site, while all other model settings were the same as those used to produce the cloudless LUT. The simulations that were performed for the altitude of the site were also used to evaluate the assumption that the UVI increases by 5% per km.

Practically there are two ways of using UVIOS2. For past data using the best available information giving priority to ground based/satellite based/modelling based data in this order of preference. For nowcasting or short term forecasting using any existing real time available data.

**Table 2. Inputs of the UVIOS2, TEMIS, and CAMS services that provide the UVI.**

| Parameter | UVIOS2 | TEMIS | CAMS |
|---|---|---|---|
| Past/reanalysis data | | | |

| Cloud inputs | Based on MSG Cloud Optical thickness | Cloud correction based on satellite data (reflectivity, cloud cover). | Dynamic cloud modeling with real-time weather forecasts. |
|---|---|---|---|
| Spatial | 5km x 5km for clouds 13 km x 24 km for Ozone | ~80 km x 40 km (GOME-2) to 13 km x 24 km (OMI). | 0.4° x 0.4° (~44 km x 44 km). |
| Temporal | Every 15 minutes | Daily updates (based on satellite overpasses). | Every 1 hour |
| Aerosol | Ground based measurements or CAMS AOD, based on availability at the location under study | Through cloud reflectivity or historical AOD | advanced atmospheric models and data assimilation from satellite and ground-based observations. |
| Total ozone | Brewer if available, mainly based on OMI | Based on OMI | Full atmospheric modeling (transport + chemistry). |
| Nowcast/forecast data | | | |
| Cloud inputs | Based on MSG Cloud Optical Thickness and cloud motion vectors (for forecast) | Not available forecasts. Only Cloudless sky UV | Dynamic cloud modeling with real-time weather forecasts. |
| Spatial | 5 km x 5 km for clouds 13 km x 24 km for Ozone | ~80 km x 40 km (GOME-2) to 13 km x 24 km (OMI). | 0.4° x 0.4° (~44 km x 44 km). |
| Temporal | Every 15 minutes up to 3 hours | Daily up to 7 days | Every 3 hours up to 5 days |
| Aerosol | Based on CAMS AOD forecasts | Historical AOD | CAMS forecasting |
| Total ozone | TEMIS forecast used (previous day) | TEMIS forecast: Based on satellite observations with some | Uses multiple satellite sources + numerical models. |

| | | basic extrapolation techniques | |
|---|---|---|---|

As shown in Table 2 there are basic differences but also common approaches in the three UVI services. The main advantage of UVIOS2 is that it provides higher spatiotemporal resolution for nowcasted or past data. Nevertheless, it utilizes CAMS and
TEMIS forecasts for AOD and ozone nowcasts/forecasts respectively. Overall, all the data used are going through libRadtran towards calculating UVI.

## 2.4 UVIOS2 inputs

Different combinations of model inputs have been used to assess the UVIOS2 accuracy when it is used for nowcasting and for climatological analyses. In all cases, the modelled cloudless-sky UVI values were derived by interpolating linearly the elements
of the 5-dimensional LUT. An overview of the data that was used to interpolate the UVI is presented in Table 3.

Default values of the aerosol asymmetry parameter (ASY) and the surface albedo were used for the simulations. Analyses of different AERONET datasets show that climatological ASY at 440 nm usually varies by about ± 0.03 around a typical value that is slightly lower than ~ 0.7 (e.g., Fountoulakis et al., 2019; Kazadzis et al., 2016; Khatri et al., 2016; Raptis et al., 2018 ). Given that ASY generally decreases with wavelength it was assumed to be 0.7 in the UV. The real ASY can however differ
occasionally by up to about ± 0.1 (e.g., Fountoulakis et al., 2019). We estimated that a difference of 0.1 in the asymmetry parameter can result in differences of up to ~ 2% in the simulated UVI. Using a default surface albedo (0.05) also introduces uncertainties in the modelling of the UV index. Surface albedo changes spectrally and its impact differs depending on aerosol load and properties (e.g., Corr et al., 2009; Fountoulakis et al., 2019). Nevertheless, during the snow-free period at Davos differences in surface albedo are estimated to be within ± 0.03 (e.g., Feister and Grewe, 1995) resulting in differences that are
of the order of a few percent. Sensitivity analysis revealed that the uncertainty in the UVI simulations for AOD ≤ 0.5 due to the combined effect of using default ASY and surface albedo values (with errors of ± 0.1 and ± 0.03 respectively) is less than 3%.

**Table 3. Combinations of input data for the UVIOS2 system for cloudless sky conditions. The three different combinations used to evaluate the system as a tool for climatological analysis are referred to as CAMS, CAMS+OMI, GB.**

| Variable | Nowcasting I (SAT) | Climatological I (CAMS) | Climatological II (CAMS+OMI) | Nowcasting II and Climatological III (GB) |
|---|---|---|---|---|
| **AOD** | CAMS forecasted | CAMS reanalysis | CAMS reanalysis AOD at 550 nm | Measured AOD at 500 nm from CIMEL |

| | AOD at 550 nm | AOD at 550 nm | | |
|---|---|---|---|---|
| **TOC** | Forecasted from TEMIS | CAMS reanalysis | OMI measured | Measured from Brewer |
| **AE** | Climatological (1.5) | Climatological (1.5) | Climatological (1.5) | Measured by CIMEL (440 – 675 nm) |

For UVI nowcasting, the aerosol properties and TOC that were used as model inputs were either forecasts (AOD, TOC) or climatological values (AE, SSA, ASY). Specifically, 1-day ahead forecasts of the TOC from TEMIS (https://www.temis.nl/uvradiation/nrt/uvindex.php) and of the AOD at 550 nm from CAMS (https://ads.atmosphere.copernicus.eu/cdsapp#!/dataset/cams-global-atmospheric-composition-forecasts?tab=overview), as well as monthly climatological values of the SSA and the AE (typical values of 0.9 and 1.5, respectively, have been estimated for Davos) were used to interpolate the elements of the LUT. Total ozone 5-days forecasts are available from TEMIS on a daily temporal resolution. Detailed description of TEMIS and the available products can be found on the service web-page (https://www.temis.nl/uvradiation/product/index.php). The CAMS forecasted AOD is available for the following 5 days, on a 1-hour resolution, and the forecasts are updated every 12 hours.

For the calculation of the climatological cloudless-sky UVI, the three combinations of inputs presented in Table 3 were used: (1) Reanalysis TOC and AOD from CAMS (Inness et al., 2019) instead of the corresponding forecasted products. All other parameters were kept the same as for nowcasting. The CAMS reanalysis, available from 2003 onwards, is the global reanalysis dataset of atmospheric composition of the European Centre for Medium-RangeWeather Forecasts (ECMWF), consisting of three-dimensional time-consistent atmospheric composition fields, including aerosols and chemical species. It is based on ECMWF's Integrated Forecast System (IFS), with several updates to the aerosol and chemistry modules described by Inness et al., (2019). CAMS reanalysis products are available from the Copernicus Atmosphere Data Store (ADS, https://ads.atmosphere.copernicus.eu/#!/home (last access on 8 August 2024)) on a 3-hourly basis on a regular 0.75° x 0.75° latitude/longitude grid (instead of their native representation). This dataset is referred to as "CAMS".

(2) TOC that has been retrieved from the Ozone Monitoring Instrument (OMI) aboard Aura (Levelt et al., 2006), and reanalysis AOD from CAMS (Inness et al., 2019). All other parameters were again kept the same as for nowcasting. This dataset is referred to as "CAMS+OMI".

(3) TOC measurements from the Brewer spectroradiometer with serial number 163 (Brewer#163) (Gröbner et al., 2021), AOD at 500 nm, and AE (440 – 675 nm) from the CIMEL radiometer (Giles et al., 2019), and all other parameters the same as for the default nowcasting setup. The AOD and AE that were used for the study are level 1.5, version 3 AERONET direct sun products. Level 2 AERONET retrievals were not used because they were not available at the time of the analysis. Since these inputs are produced in near real time, we consider that they could be potentially used for UVI nowcasting in addition to

climatological studies. Level 2 AERONET retrievals are available with a longer latency and can be used for reanalysis at a later stage. This dataset is referred to as "GB".

The cloudless-sky UVI LUT outputs were in all cases post corrected for the effect of the varying Earth-Sun distance and for the surface elevation (1596 m for Davos). The all-sky UVI values were derived in all cases by multiplying the cloudless-sky UVI with the Cloud Modification Factor in UV (CMFUV), which was calculated as described in Sect. 2.1 from the MSG-SEVIRI COT.

The UVI was simulated for the period 1 July – 20 August 2022 at the time of the QASUMEII measurements (15 min temporal
resolution). The MSG images, and thus the CMFUV, were available at the exact time of the UV scans. All the other parameters (AOD, TOC, etc) were interpolated linearly to the time of the measurements.

For the analysis, measurements were classified as clear-sky (i.e., sun was not fully or partially covered by clouds) and all-sky (i.e., for all cloudiness conditions). In the following, clear-sky conditions refer to unoccluded solar disc according to measurements (although clouds may be present on the sky). Cloudless-sky conditions refer to cloud-free skies. To classify the
measurements, the direct component of the total solar irradiance, as it was measured by the pyrheliometer that was operating at Davos during the campaign, was simulated as described in Papachristopoulou et al. (2024), and was then compared to the measured direct irradiance. When differences between the two components exceeded 10%, we considered that the sun was (fully or partially) covered by clouds.

## 3 Results

### 3.1 Assessment of UVIOS2 for real time applications

In this section we tried to assess the accuracy of the modelled UVI when UVIOS2 is used for real time applications. Initially we compared the modelled and the measured UVI under clear-sky and all-sky conditions. The UVI was modelled using the default inputs and setup of the UVIOS2 (SAT), as well as using high quality ground-based measurements (GB), that theoretically can be available at near real time for the retrieval of a higher accuracy estimate of the UVI.

### 3.1.1 Clear-sky UVI

Under clear-skies, the ratio between modelled UVI datasets and the corresponding measured UVI from QASUMEII was then calculated and the results are shown in Fig. 2.

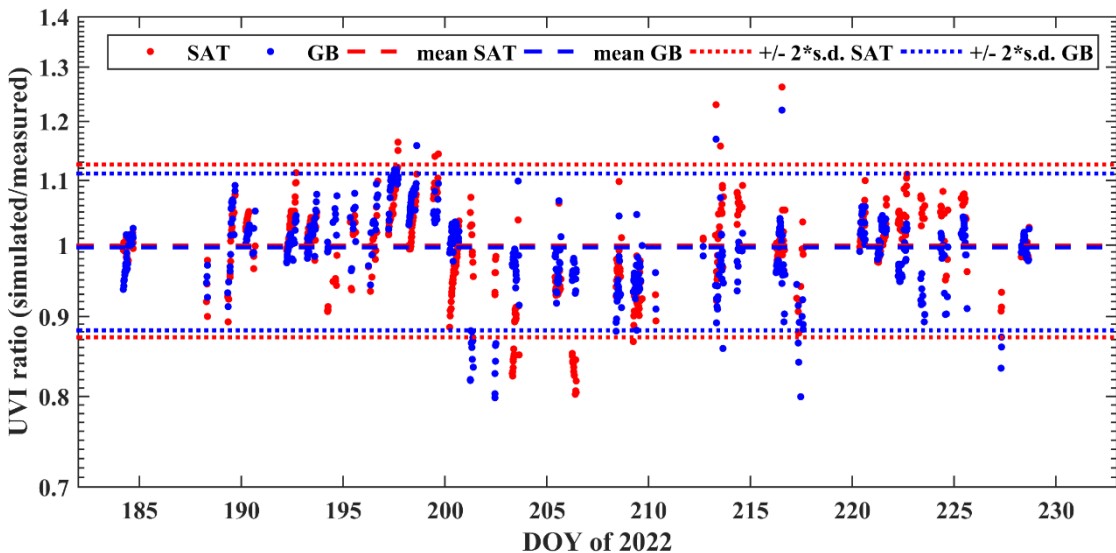


**Figure 2. Ratios between simulated and measured clear-sky UVI.** Red colour: ratios for simulations performed using forecasted CAMS AOD and TEMIS TOC. Blue colour: ratios for simulations performed using measured AOD, AE (by CIMEL), and TOC (by Brewer). Ratios have been calculated for SZA < 75°. Dashed lines represent the mean, while dotted lines represent the range of 2

standard deviation.

While the average ratio is in both cases ~0.99, the standard deviation is high, 0.063 and 0.057 for SAT and GB respectively, i.e., only slightly lower for GB. This result shows that using highly accurate inputs for TOC, AOD at 500 nm, and AE does not result in a noticeable improvement in the accuracy of the average modelled clear-sky UVI (standard deviation decreases by only a few percent), which means that other factors are also important for the calculation of the surface UVI levels at Davos.

The role of each of the factors that were found to be the most important is discussed in the following.

**AOD and TOC:** The AOD forecasted by CAMS is at a different wavelength (550 nm) relative to the AOD measured by the CIMEL (500 nm). To compare the AOD from the two different sources, the AOD from the CIMEL was extrapolated at 550 nm using the measured AE (440 – 675 nm). The differences between the AOD at 550 nm from the CIMEL and CAMS are shown in the Appendix (Figure A1). Differences in AOD are in most cases within ± 0.1, with an average of ~0, which explains

differences of up to about ±10% between the UVIs simulated using the two different datasets. When differences in AOD are larger (e.g., in day of the year (DOY) 201 – 202 CAMS has not captured the large AOD levels over the site and the AOD from CAMS is lower by 0.15 – 0.25 relative to the AOD from CIMEL) they result in correspondingly larger differences between the ratios (of 10 – 20%). Differences in TOC (Figure A2 in the Appendix) are generally within ± 25 DU, with an average of about 4 DU (on average, TEMIS slightly overestimates TOC for the period of the campaign), but occasionally they can reach

± 40 DU. Differences of ± 25 DU in TOC can explain differences of about ± 15% in the UVI modelled using the two different datasets (e.g., Kim et al., 2013). The large differences between the ratios that were calculated for the two different UVI datasets in DOY 194 are mostly explained by differences in TOC (~20 DU during most of the day). Differences in DOYs 200 – 204,

206, and 223 are mostly explained by differences in AOD. The accuracy in the ground-based measurements (~ 0.02 for the AOD (Giles et al., 2019) and better than 2.5% for TOC (e.g., Carlund et al., 2017; Fountoulakis et al., 2019) cannot explain the standard deviation of 0.057 in the ratio between the modelled UVI when GB measurements are used as inputs and the measured UVI.

**SSA:** While a default SSA value of 0.9 has been used for the simulations, the real SSA at the shorter UV wavelengths, which contribute the most in UVI, can differ significantly, ranging from values smaller than 0.8 (during e.g., events of dust, pollution or biomass burning aerosols that have been transferred over the site) to values exceeding 0.98 (e.g., for mixtures that are dominated by sulfuric aerosols). As discussed in Krotkov et al., (1998) the SSA has a very significant impact on the UVI. In their study they show that assuming very absorbing aerosols (SSA = 0.6) results in about half of the UVI for highly reflective aerosols (SSA = 0.99) for AOD =1 at 325 nm. Confirming the findings of Krotkov et al., (1998), in Figure 3 we show that the sensitivity of the ratios to the input SSA increases with increasing AOD. For AOD between 0.3 and 0.4 a change of 0.1 (increase or decrease) in SSA results in a change of ~ 0.1 in the ratio (i.e., ~10% in the simulated UVI) which is of similar magnitude with the change in UVI due to a change of ~ 0.1 in AOD. The effect of changing SSA becomes less significant as the AOD decreases. Nevertheless, even for AOD of ~ 0.1, a change of ~ 0.1 in the SSA results in a change of 0.05 in the ratio (i.e., of ~ 5% in the modelled UVI). Generally, Figure 3 denotes that aerosol mixtures over Davos in the summer are dominated by aerosols that are weak absorbers of the UV radiation.

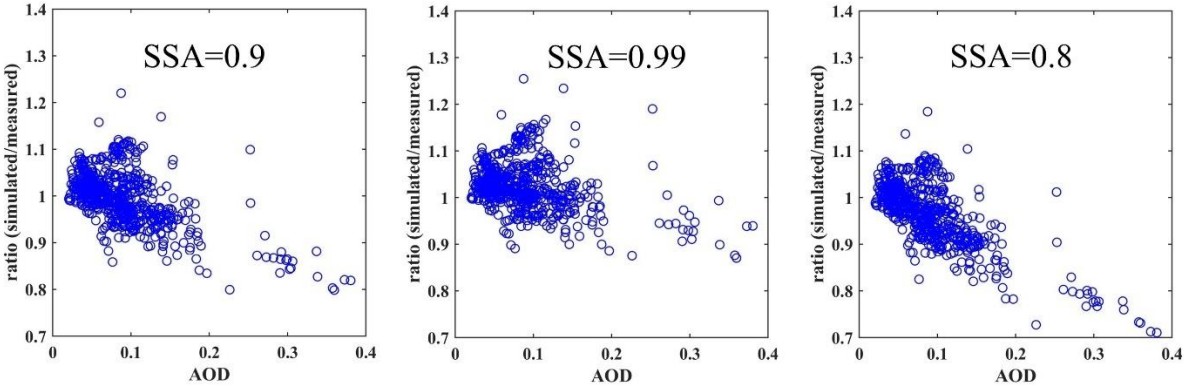

**Figure 3. Ratios between simulated (using GB measurements) and measured clear-sky UVI when different SSA values are used for the simulations.**

Changing the SSA from 0.8 to 0.99 results in mean ratio values that are similar to each other and close to unity (see Figure A3 in the appendix). Using a similar analysis, Krotkov et al., (1998) concluded that the SSA that gave the smallest gradient with AOD is more representative for Toronto. However, in our case, the analysis of the SSA at 440 nm from AERONET for the period of the campaign results in an average value below 0.95, and since for more aerosol species the SSA increases with decreasing wavelength, we estimate that an SSA equal to 0.9 is more appropriate to model the UVI at Davos.

The high values of the ratios between modelled and measured UVIs in DOY 197 – 199 can be possibly justified by real SSA values that are lower than 0.9, and thus assuming SSA=0.9 for the simulations results in an overestimation of the UVI. In these days, the SSA at 440 nm from AERONET was generally lower than 0.9 (values between 0.77 and 0.92). As shown in Figure 4, the low SSA values may be due to polluted air masses originating from low altitudes over Germany. During DOY 200 – 210 when a negative bias is evident in Fig. 2, in addition to the broken cloud conditions that occasionally enhanced significantly the real UVI (see the discussion below), high levels of scattering aerosols were recorded over Davos (AOD at 340 nm from AERONET was ~ 0.6 on DOY 201-202, and SSA at 440 nm was generally above 0.97). We were not able to identify the conditions that favoured the presence of such high loads of reflective aerosols at the region. On DOY 201 and 202 the AOD is underestimated by CAMS by up to 0.25 (Fig. A1). Nevertheless, on these days the agreement between the measured and modelled UVI is much better for SAT (simulated using CAMS AOD) relative to GB. By comparing CIMEL AOD measurements with measurements from other photometers we confirmed that they are accurate. As shown in Fig. 5, broken cloud conditions during these days cannot explain the UVI enhancement (as e.g., in DOY 217). When the CAMS AOD data were used to simulate the UVI, errors in the AOD and the AE possibly counterbalanced the large errors due to the SSA which possibly resulted in better agreement between the modelled and the measured UVI for the particular dataset (see Fig. 2).

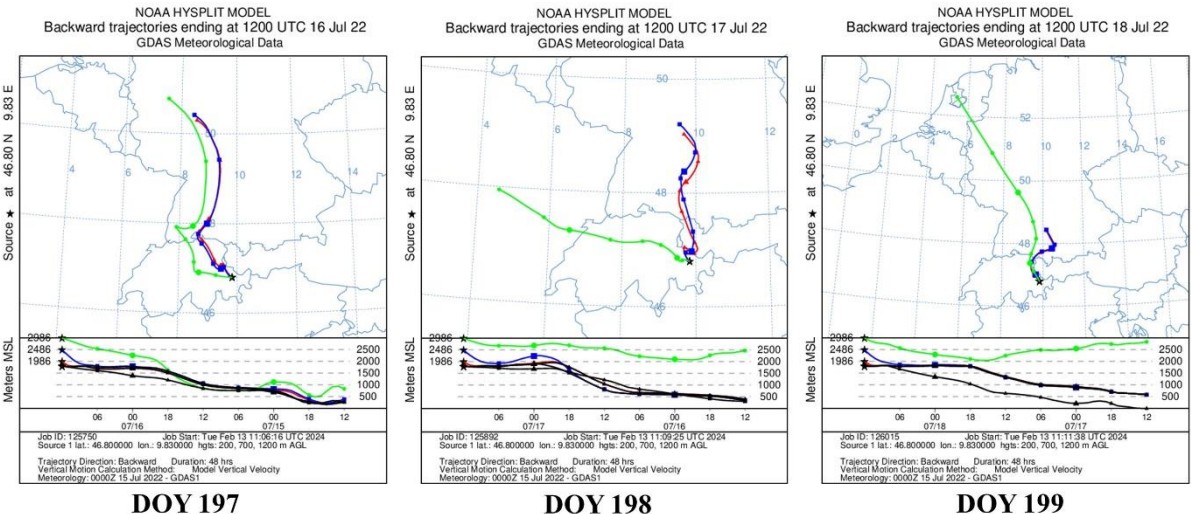

**Figure 4. HYSPLIT back-trajectories of the air masses that arrived over Davos (at altitudes 400, 900, and 1400 m over the site).**

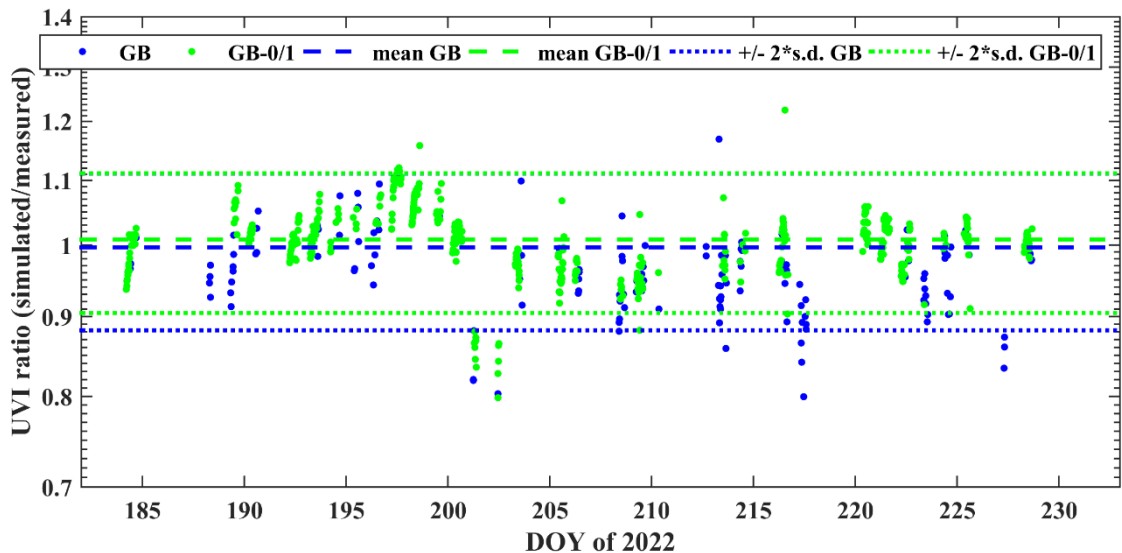

**Figure 5. Ratios between simulated and measured clear-sky UVI for simulations performed using measured AOD, AE (by CIMEL), and TOC (by Brewer). Blue color: Unoccluded sun with cloudiness in the sky between 0 and 7 octas. Green color: Unoccluded sun with cloudiness in the sky between 0 and 1 octas.**

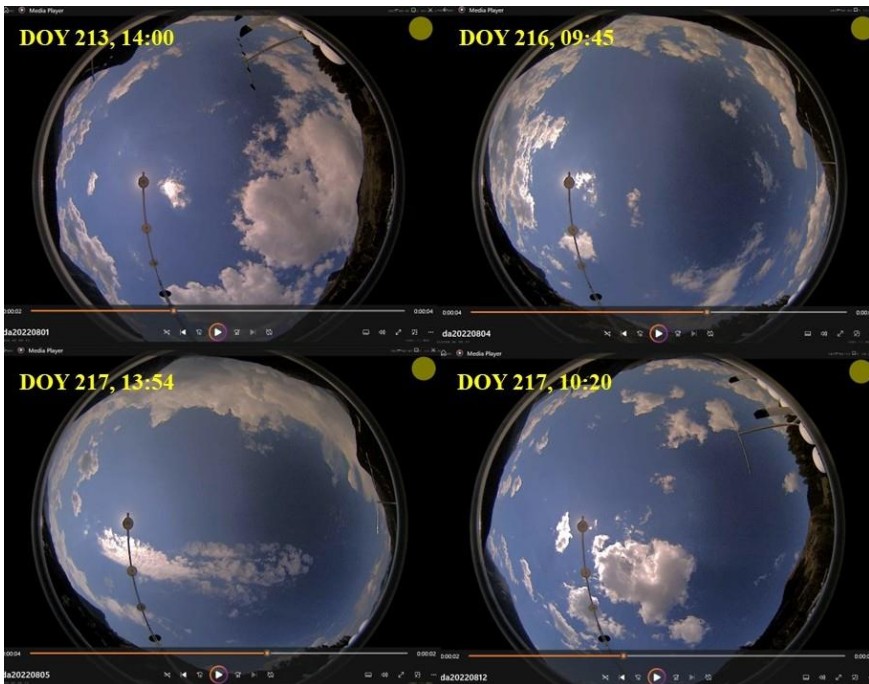

**Figure 6.** Sky camera images for different DOY and times, which show the clouds near the sun that enhance the UVI at the surface.


**Clouds near the sun that do not cover the solar disc:** At high altitude stations such as Davos, the presence of orographic clouds is frequent (some examples are shown in Fig. 6). Such clouds can enhance the UVI at the surface if they do not cover the solar disc, by redirecting part of the diffuse irradiance to the surface. Using the GB UVI dataset we tried to assess the impact of the presence of clouds when the sun is unoccluded. By comparing the ratios when there are very few or no clouds

in the sky (unoccluded sun, 0 or 1 octas, green points in Fig. 5) with the ratios for all cloudiness conditions (unoccluded sun, 0 - 7 octas, blue points in Fig. 6) we confirm that including conditions with 2-7 octas adds a small negative bias to the ratio, possibly due to the enhancement by clouds. Removing these cases slightly improves the ratio and reduces the variability.

**Other factors of uncertainty in the modelling of the clear-sky UVI:** As discussed earlier, fixed values of the ASY, and the surface albedo have been used to derive the clear-sky UVI. The ASY has been estimated using Kinne, (2019) climatology.

Although ASY may deviate from the typical value of 0.7 depending on the type of aerosols, the uncertainties related to ASY are minor relative to the overall simulation uncertainties, as already discussed. The fixed value of 0.05 for surface albedo is also not expected to induce uncertainties larger than 2% in the simulations (e.g., Fountoulakis et al., 2019). The standard correction (of 5%/km, i.e., ~8% for Davos) for the effect of altitude also induces small uncertainties (e.g. Blumthaler et al., 1997; Chubarova et al., 2016). We compared the UVI that was simulated with UVSPEC for the conditions of the campaign

and for the altitude of the site with the results that were derived using the LUT (see Section 3.1.2), and that were post-corrected for the effect of altitude, and we found differences that were generally smaller than 1%. The horizon in Davos is limited by the tall mountains surrounding the site, which at SZAs larger than 75° can block the direct component of the solar irradiance, as well as a large fraction of the diffuse irradiance (Hülsen et al., 2020). Although we have not corrected the modelled UVI for the effect of limited horizon, for SZA<75°, this effect combined with the effect of default altitude correction was estimated

to induce uncertainties smaller than 2%. Considering invariant atmospheric properties (i.e., pressure and temperature profiles) based on a standard atmospheric profile (Anderson et al., 1986) which is not necessarily representative for a mountainous site such as Davos, introduces additional uncertainty, which however is expected to be minor relative to the overall uncertainty budget in our estimates. The used ETS and ozone absorption cross sections are more significant uncertainty factors (see Section 2.2).

**3.1.2 All-skies UVI**

For the default UVIOS2 setup the clear-sky UVI (that is derived using AOD from CAMS and TOC from TEMIS) is multiplied with the CMFUV to calculate the all-sky UVI. This method has been preferred instead of performing directly libRadtran simulations using cloud optical properties as libRadtran inputs because it is much faster and has a minor impact on the simulated UVI uncertainty compared to the uncertainty induced by the assumption of cloud homogeneity in the satellite pixel

(e.g., Schenzinger et al., 2023). The all-sky UVI that was calculated using the LUT, the all-sky UVI that was calculated directly

from libRadtran (for the altitude of the station and using cloud optical properties as inputs), and the UVI measured by QASUMEII are shown in Fig. 7 for DOY 188 - 191. During the cloudy DOY 188 and 191 the variability in the modelled UVI (using both approaches) is in quite good agreement with the variability in the measured UVI. As shown in Fig. 8, both approaches result in correlation coefficients of ~ 0.95 between the measured and modelled UVIs. The differences between the
measured and modelled UVIs (for SZAs below 75°) are presented in Fig. 9. For both modelling approaches the average differences in the UVI are nearly identical (~ 0) with a nearly identical standard deviation (~ 1). From figures 7 - 9 it is obvious that the differences between the two modelling approaches are very small, and that the deviations originate mostly from the assumption of homogeneous distribution of clouds within the satellite pixel. The differences between the UVI from QASUMEII and the model (with both setups) are in some cases very large, reaching even values of ± 8. These large differences
are mainly due to the model inability to predict accurately if the fraction of the solar disc is occluded by clouds, especially under broken cloud conditions.

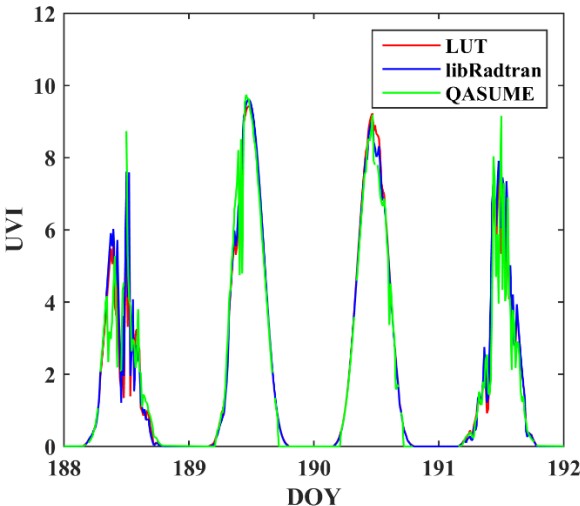

**Figure 7. Measured UVI, and modelled UVI from the LUT and libRadtran.**

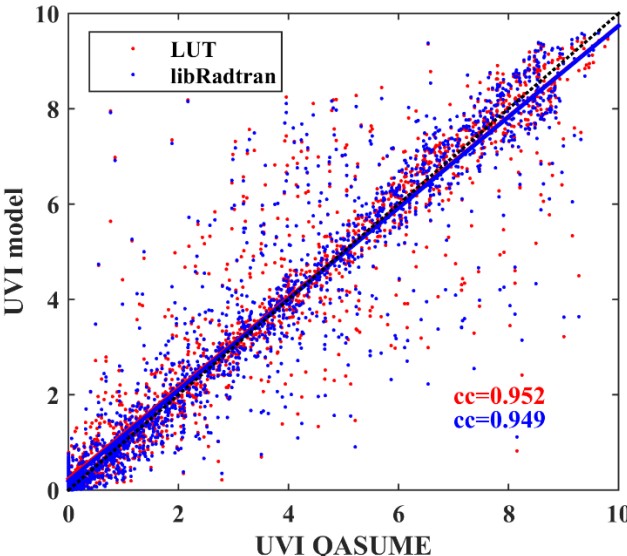

**Figure 8. Correlation between measured UVI and UVI that was modelled using the two different approaches.**


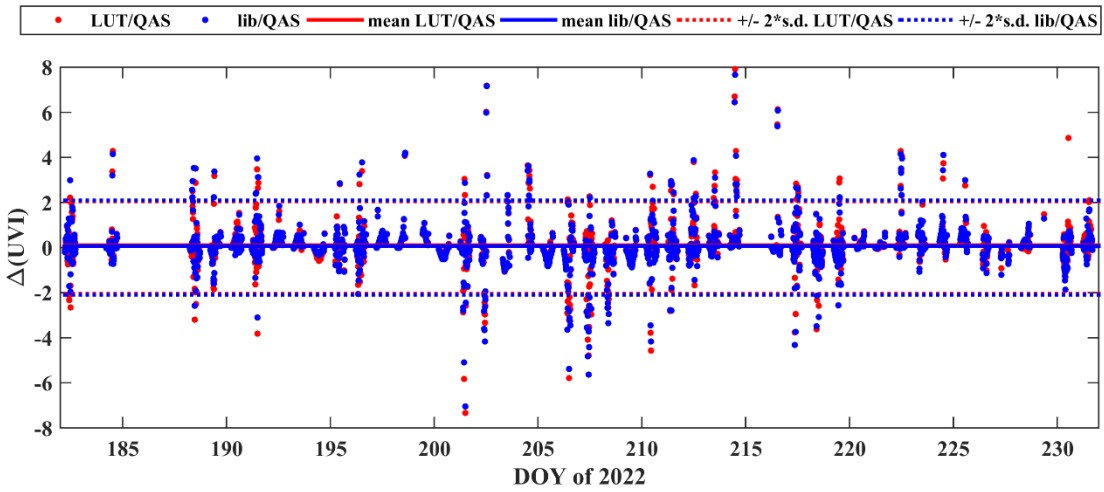

**Figure 9. Differences (QASUMEII-model) between measured and modelled UVI**

## 3.2 Assessment of UVIOS2 for climatological studies

In this section we assess the accuracy of the modelled UVI based either on AOD and TOC measured at Davos or taken from reanalysis data are used as inputs for the UVIOS2 system (instead of real time measurements or forecasts), that is the case when the system is used for climatological studies. The results of the comparison between the UVI from the QASUMEII and the UVI for the three different input datasets that were used as libRadtran inputs (i.e., ground-based TOC and AOD (GB), CAMS reanalysis AOD and TOC (CAMS), CAMS reanalysis AOD and OMI TOC (CAMS+OMI)) are shown in Fig. 10. The comparison has been performed again for SZA<75°. In all cases, the average agreement between the measured and the modelled UVI is nearly perfect (average differences of ~ 0). The standard deviation (2 x standard deviation ~1.9) is very similar to the standard deviation that was calculated for the UVI that was modelled using forecasted CAMS inputs. For the UVI that was modelled using GB data the standard deviation is slightly lower (2 x standard deviation ~1.7).

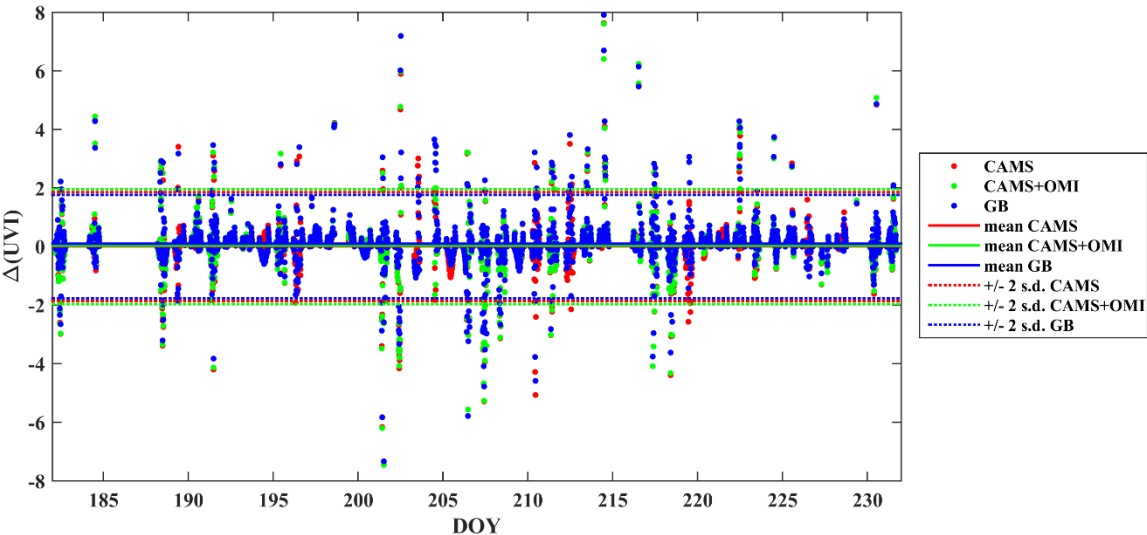

**Figure 10. Differences between the measured UVI and the modelled UVI for different UVIOS2 inputs.**

A summary of the results from the comparison between UVIOS2 and QASUMEII for the most uncertain (Nowcasting I - SAT) and the most accurate (Climatological III - GB) setup is presented in Table 4. The results for the two other setups (Climatological I – CAMS, Climatological II – CAMS+OMI) are very similar to those shown in Table 4, and thus they are not presented here.

**Table 4. Median differences and the corresponding standard deviation between UVIOS2 and QSUMEII for SZA < 75°. Columns 2–5 are statistics based on the absolute UVI value while columns 6–9 are statistics for relative differences.**

| Setup | All sky $\Delta$(UVI) | St. d. | Clear sky $\Delta$(UVI) | St. d. | All sky (%) | St. d. | Clear sky (%) | St. d. |
|---|---|---|---|---|---|---|---|---|
| **Nowcasting I (SAT)** | 0.05 | 1.04 | 0.01 | 0.35 | 1.8 % | > 100 % | 0.4 % | 6.3 % |
| **Nowcasting II and Climatological III (GB)** | -0.05 | 1.02 | 0.00 | 0.34 | 1.8 % | > 100 % | 0.1 % | 5.7 % |

### 3.3 Comparison with the results of the campaign

In this section we tried to assess the performance of UVIOS2 (with the default setup, i.e., forecasted AOD and TOC from CAMS and TEMIS respectively) with respect to the accuracy of the filter radiometers that participated in the campaign. The relative % differences between the UVI from the radiometers and QASUMEII (100% x (radiometer-QASUMEII)/QASUMEII) are presented for two cases: (1) when the calibration provided by the operator is used (USER) and (2) when the PMOD/WRC calibration is used (PMOD/WRC). In this section there is no extensive discussion relative to the results of the comparison between QASUMEII and the radiometers, since they have been already discussed thoroughly in Hülsen and Gröbner, (2023). The median of differences between UVIs from QASUMEII and UVIOS2 is less than 2%, which confirms that UVIOS2 simulated very accurately the average UVI levels over Davos during the period of the campaign, as was also discussed in sections 3.1 and 3.2. Nevertheless, the 2.5[th] and 97.5[th] percentiles are at about -20% and 35% respectively. As shown in the previous sections this large spread is mostly due to the inability of the model to predict accurately solar disc occlusion, especially under broken cloud conditions.

When the USER calibration is used to derive the UVI from the radiometers, the range between the 2.5[th] - 97.5[th] percentile is in some cases of the order of 40%, and thus comparable to the corresponding range for UVIOS2 (~ 55%). When the PMOD/WRC calibration is used, the range between the 2.5[th] and 97.5[th] percentiles is in all cases below 20%. These latter findings show the significance of the systematic maintenance and calibration of the sensors.

From Fig. 11 it is clear that, although UVIOS2 achieves to simulate very accurately the average UVI levels, it is still significantly more uncertain than the measurements from standard UV radiometers when they are well calibrated and well maintained, especially under broken cloud conditions.

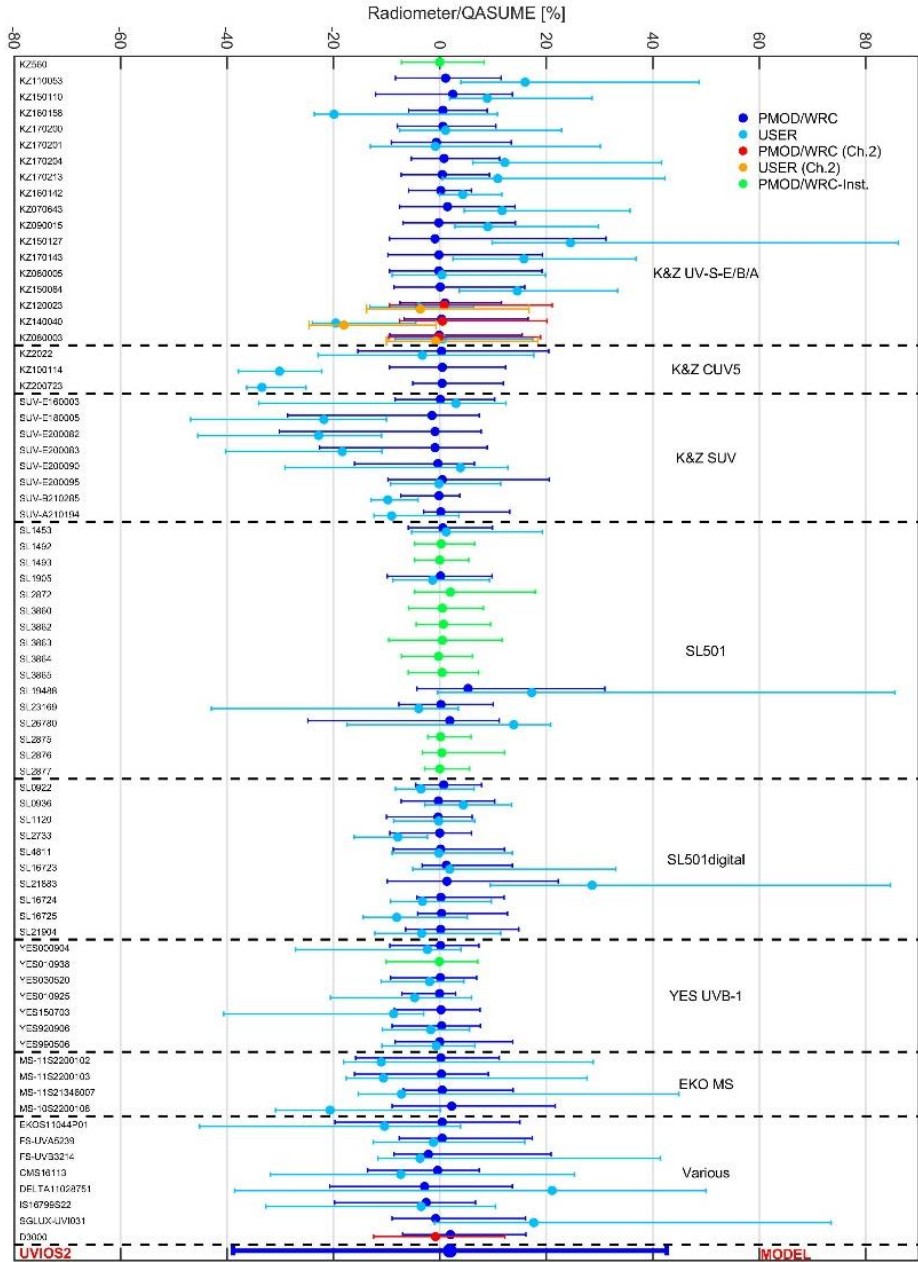

**Figure 11. Relative (in %) difference between the UVI measured by QASUME and by the filter radiometers using the operator (USER) and the new (PMOD/WRC) calibration factors. The results for the UV-A channel (ch. 2) are also shown for dual channel radiometers. Green dots represent instruments that operate regularly at PMOD/WRC. Comparison has been also performed with the UVI from UVIOS2 (SAT configuration). Displayed is the median of the ratio for the calibration period as well as the 2.5th and 97.5th percentile for each instrument (95% coverage). Figure has been originally published in** (Hülsen and Gröbner, 2023)**.**

## 4 Summary and conclusions

The UVIOS2 is an upgrade of the UVIOS system described in Kosmopoulos et al. (2021). We described the new features of the system, and evaluated in depth its performance at Davos, Switzerland during the UVCIII campaign. We assessed the accuracy of the UVIOS2 system when it is used for UVI forecasting and when it is used for climatological studies. To achieve that, the system outputs during the UVCIII campaign were compared to the measurements of the world reference QASUME, as well as to the measurements of the filter radiometers that participated to the campaign. The performance of the system was found to be excellent in simulating the average UVI levels, but uncertainties are larger in the simulations of the instantaneous UVI values.

Our analysis showed that the parameterization used to derive CMFUV works quite well and has reduced processing time significantly without affecting the accuracy in the system outputs. This is in agreement with the findings of Papachristopoulou et al. (2024) who have shown that the parameterization used to derive the CMF (for total solar irradiance) from COT works well. Further investigation is needed to assess the uncertainties due to the cloud effect parameterization over higher albedo terrains, although we expect that the assumption of homogeneity in the satellite pixel would still be the main uncertainty factor (e.g., large gradients in cloudiness, aerosol properties, surface albedo, surface altitude).

The satellite algorithm considers that each pixel is homogeneously covered by clouds, and thus it cannot accurately determine under inhomogeneous cloudiness conditions whether the solar disc is occluded or unoccluded. Furthermore, when the solar disc is occluded, there are uncertainties in the retrieved COT, and even larger uncertainties in the CMFUV that is subsequently calculated. The above can result in very large (especially %) differences between the measurements and the simulations. When the sun is unoccluded, broken clouds in the sky can contribute to the enhancement of the UVI at the surface.

The differences between the measured, forecasted, and reanalysed AOD and TOC data that were used as inputs are not critical for the accuracy of the UVIOS2 outputs for Davos, and thus the performance of UVIOS2 does not differ significantly when it is used as a forecasting tool or as a tool for climatological studies. Under cloudless conditions the role of SSA was found to be equally important to the role of AOD, even at a (usually) low aerosol mountainous site such as Davos.

The differences between measured and modelled UVI have been also discussed in previous studies (e.g., De Backer et al., 2001; Fioletov et al., 2004; Kylling et al., 1997; Mayer et al., 1997; Reuder and Schwander, 1999; Weihs and Webb, 1997), which have also shown that the assumptions relative to the aerosol optical properties constitute a major uncertainty factor under cloudless sky conditions. Uncertainties of 5–8% at 380 nm and even larger at 305 nm were associated with the assumptions related to the aerosol optical properties. The significant role of clouds in the UVI forecasting has been also discussed in Malinovic-Milicevic and Mihailovic (2011), who used a numerical model to estimate the UVI at Vojvodina region (Serbia). Vitt et al. (2020) constructed UVI maps for whole Europe based on monthly means and showed that uncertainties in surface albedo can be also significant over mountainous mid-latitude sites in winter. Dahlback and Stamnes (1991) reported that at the Tibetan Plateau (3000-5000 m a.s.l.) the UVI can be enhanced by ~30% under broken cloud conditions compared to cloudless sky conditions. Similar impacts by clouds on the UVI were reported by Allaart et al. (2004).

Our study makes clear that there is a need for more accurate representation of the SSA in UV models in order to achieve more accurate modelling under cloudless conditions, , even for high altitude sites where the effect of aerosols is usually considered small. When AOD (at 500 nm) is 0.3 – 0.4, the effect of changes in the AOD is similar with the effect of changes of the same magnitude in the SSA. SSA measurements at shorter, and more effective regarding their biological impacts, UV wavelengths are not easy to perform, and long-time series are not available  (e.g., Bais et al., 2019; Corr et al., 2009; Go et al., 2020; Mok et al., 2016, 2018). Capturing the instantaneous impact of clouds using satellite images is impossible, especially for enhancement events, since satellite derived CMF represents spatially averaged cloud transmittance over several kilometers, while ground instrument measures instantaneous irradiance at a given location. Time averaging of the ground measurements can only partly mitigate this difference. In this study we showed that enhancement events can not be easily captured even when ground-based information is used. Since using satellite-based cloud information is the only way to forecast the UVI at large spatial scales, the accuracy in the description of the effects of clouds is a common problem for UV models that cannot be easily solved.

**Acknowledgments**

This work was supported by computational time granted by the National Infrastructures for Research and Technology S.A. (GRNET) at the National HPC facility – ARIS – under project ID pa210301-SO-LISIS. The European Commission project "EXCELSIOR": ERATOSTHENES: Excellence Research Centre for Earth Surveillance and Space-Based Monitoring of the Environment (grant no. 857510) is also acknowledged. C. S. Zerefos would like to acknowledge CAMS2_82 Project: Evaluation and Quality Control (EQC) of global products. D. Kouklaki would like to acknowledge the PANGEA4CalVal project (Grant Agreement 101079201) funded by the European Union.

S. Kazadzis acknowledges ACTRIS-CH (Aerosol, Clouds and Trace Gases Research Infrastructure – Swiss contribution), funded by the State Secretariat for Education, Research and Innovation. SK, IF, KP would like to acknowledge HARMONIA (International network for harmonization of atmospheric aerosol retrievals from ground-based photometers; grant no. CA21119), supported by COST (European Cooperation in Science and Technology).

**Financial Support**

This project has received funding from the European Union's Horizon 2020 research and innovation programme EIFFEL under grant agreement No 101003518.

**Code and Data availability**

Analytical description and instructions on how to access the model are provided in Kosmopoulos et al., (2021). All codes and datasets that are necessary to reproduce the results used in this paper are archived on Zenodo (Fountoulakis, 2025):

https://doi.org/10.5281/zenodo.16781118.

**Competing interests**

The contact author has declared that none of the authors has any competing interests.

**Author contribution**

Conceptualization: SK, IF. Methodology: SK, IF and KP. Formal analysis: IF, KP, JG, GH, and DK. Software: IF, KP, and I-

585 PR. Validation: IF, KP, SK, GH, and JG. Investigation: IF and KP. Resources: SK and CK. Data curation: IF, KP, NK, JG and GH. Visualization: IF, KP and GH. Writing (original draft preparation): IF, KP, and AM. Supervision: SK. Writing (review and editing): IF, KP, SK, JG, GH, I-PR, DK, AM, CK, and CZ. All authors gave final approval for publication.

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

**Appendix**

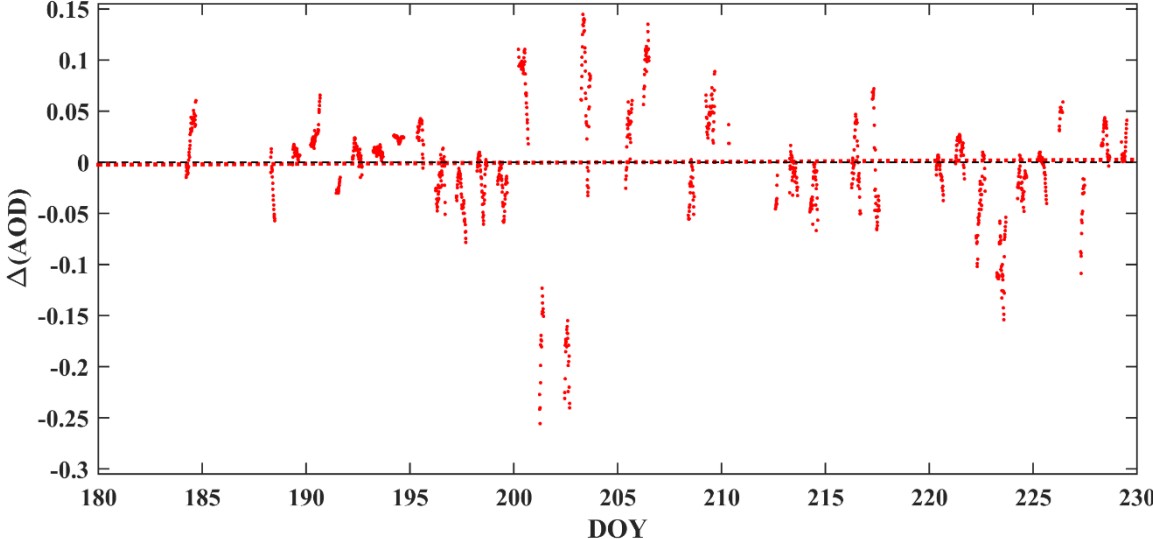

**Figure A1: Differences between the 550 nm AOD from CAMS and the CIMEL (CAMS-CIMEL).**

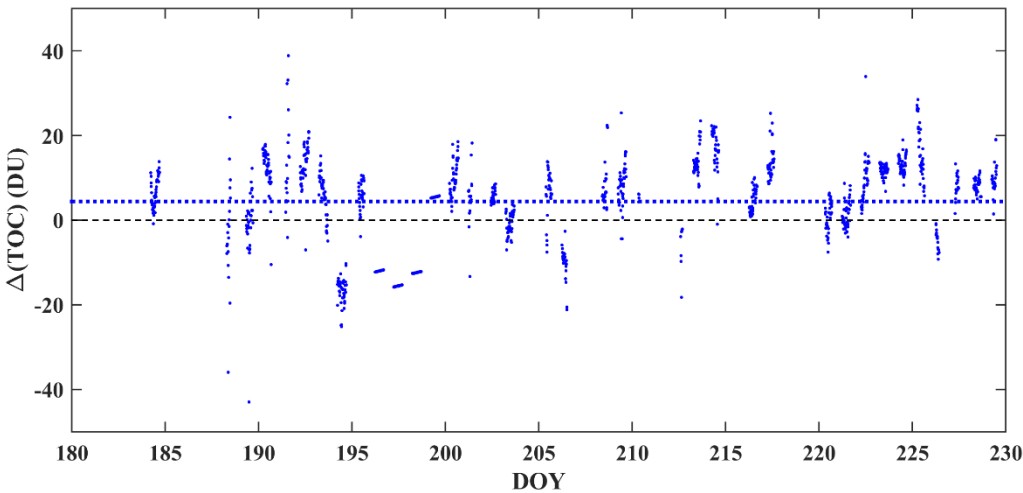

**Figure A2: Differences between the TOC from TEMIS and the Brewer (TEMIS-Brewer).**

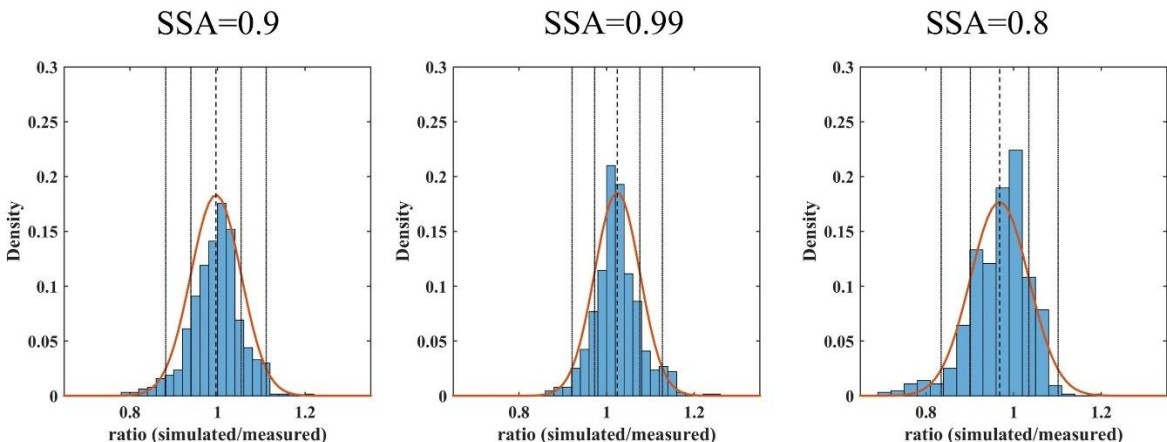


**Figure A3: Density plots of the ratio between simulated (using GB measurements) and measured clear-sky UVI when different SSA values are used for the simulations. Vertical lines represent the mean and the 1-sigma and 2-sigma intervals if a normal distribution (red line) is assumed.**