# Peer review of "Assessment of the accuracy in UV index modelling using the UVIOS2 system during the UVC-III campaign"

_EGUsphere, 2024_

## Author Comment (AC2)

We thank the reviewer for his/her analytical comments that helped us to improve the manuscript. Replies to the reviewer's comments are provided below. Line numbers refer to the version with track changes.

**GENERAL COMMENTS**

The manuscript by Fountoulakis et al. presents the results of a modelling system for estimating the UV Index at the surface (UVIOS2) during an international comparison campaign of broadband UV radiometers in Davos. The system is described, and the results are illustrated for both (i) clear-sky days and (ii) all-sky conditions. The factors influencing the comparison are discussed.

In my opinion, the research topic is important and within the scope of the journal, and the results are in good agreement with the reference instrument. However, the description of the methodology and the presentation of the results could be improved. Hence, I recommend publication after the authors address the issues outlined below.

**SPECIFIC COMMENTS**

- The study evaluates the performance of the UVIOS2 system during a summer campaign (June–August). The authors should be cautious not to extrapolate these results to seasons when snow is present near the measurement area, as surface albedo determination and the estimation of cloud properties from satellite data could be problematic. Additionally, at higher altitudes and latitudes, snow or ice may persist even in summer. A discussion on this topic would be beneficial.

**Reply**

Discussion about the role of surface albedo has been also provided in the introduction. The following lines have been added (103-107):

"In general, the change in the levels of the solar UV irradiance with altitude depends on atmospheric composition and has a strong wavelength dependence which is introducing difficulties in the modelling of the UVI at mountainous sites (e.g, (Dvorkin and Steinberger, 1999; Krotkov et al., 1998)). At very high altitude (or/and latitude) sites ice and/or snow may persist even in late spring and summer resulting in extremely high UV exposure for skiers or other visitors (e.g., (Schmalwieser et al., 2017b; Siani et al., 2008; Utrillas et al., 2016))."

In lines 143-144 at the introduction, we also added the following:

"It must be clarified that the study refers to a snow-free period at Davos, and thus the uncertainties related to the parameterization of surface albedo, which may be significant for higher altitude sites even in the summer, are not quantified or discussed here."

To further clarify what we mean with "Further investigation is needed to assess the uncertainties due to the cloud effect parameterization over higher albedo terrains, although we expect that the assumption of homogeneity in the satellite pixel would still be the main uncertainty factor"

**A brief reference is given in lines 423–424 ("we expect that the assumption of homogeneity in the satellite pixel would still be the main uncertainty factor"), but the basis for this expectation is unclear.**

**Reply**

We added at the end of the sentence (line 537):

 "(e.g., large gradients in cloudiness, aerosol properties, surface albedo, surface altitude)."

**Could the authors provide a bibliographic reference? Moreover, depending on the SEVIRI channel used for retrieving the COT, is there sufficient contrast between snow and clouds?**

**Reply**

Since we do not refer to snow/ice-covered terrain, we did not add any discussion in the manuscript relative to the COT retrieval from SEVIRI, and the contrast between snow and clouds. We provide the required information here

The retrieval of COT (by the CMIC algorithm) is performed only for satellite pixels that have been caracterized as cloudy by the Cloud Mask (CMA) product, which distinguishes between snow/ice covered areas and clouds (Météo-France, 2016). Cloud free areas covered by snow or ice are identified during the daytime by their very low reflectivity at 1.6 or 3.8 μm and high reflectivity at 0.6 μm. Moreover, thin cirrus clouds can be detected and distinguished from snow and ice due to their different emissivities at 10.8 micron and 12.0 μm. Thus, there are sufficient thresholds to differentiate between snow and clouds. The validation of the CMA product shows that the accuracy of cloud detection reaches up to 97.1% over European areas using SYNOP observations (Meteo France, 2016).

- **Role of broadband radiometers in the manuscript: The authors state in the abstract that "Radiometers that were not properly maintained and/or calibrated were found to provide UVI measurements with uncertainty that was comparable to the uncertainty of the UVIOS2 estimates, which highlights the significance of systematic maintenance and calibration of the UV**

**radiometers."** They further state (lines 434–436) that "The uncertainty in the UVIOS2 forecasts was found to be comparable to the measurements of filter radiometers when they were not properly calibrated, but ~3 times larger compared to the measurements of accurately calibrated radiometers, which shows the significance of systematic and accurate calibration of such instruments." However, is a model really needed to affirm the importance of systematic and accurate calibration of UV instruments? The comparison of the model accuracy with that of the least reliable broadband radiometers is not particularly insightful. For example, should the conclusion be that UVIOS2 estimates are only as accurate as an uncalibrated or poorly maintained radiometer? In summary, are considerations regarding broadband radiometers necessary in this paper?

**Reply**

The reviewer is right. We have removed the considerations regarding broadband radiometers from the paper.

- **Model inputs require a more detailed description:**

1. **Extraterrestrial spectrum: A more recent extraterrestrial spectrum (QASUMEFTS) exists compared to "Atlas plus MODTRAN." Given that QASUMEFTS was obtained using the same instrument employed here as a reference, why was it not used for the calculations? What is the expected deviation resulting from this choice?**

**Reply**

The pre-calculated LUTs (i.e., the "core" of UVIOS2) were based on the Atlas Plus MODTRAN ETS. Using a different ETS would require either recalculation of the LUTs or to run libRadtran directly. Nevertheless, we added the following lines in the document (lines 214 - 216):

"Using a different ETS might result to differences in the simulated erythemal irradiances, as for example was shown in the study of (Gröbner et al., 2017). Based on the results of the latter study we estimate that the simulated irradiances might differ by up to 5% if a different ETS was used, making the used ETS spectrum a major uncertainty factor in UVIOS II simulations. "

2. **Cross-sections: These should be presented in a separate table. Are the ozone cross-sections used in the model consistent with Brewer retrievals?**

**Reply**

We do not believe that a separate table is necessary since the appropriate reference is provided. Nevertheless, we have added the following lines (lines 218 - 223):

"TOC is among the main regulators for the UVI levels at the surface and thus using TOC values that have been retrieved using different ozone absorption cross sections relative to those that have been used to create the LUT (Molina and Molina, 1986) would result in differences between the measured and the simulated UVI. Differences of 1 - 3% have been reported in the retrieved TOC depending on the used absorption cross sections (Fragkos et al., 2015; Redondas et al., 2014), which may result in differences of up to ~5% in the calculated UVI, depending mainly on the used cross sections, the SZA, aerosols load, and cloudiness (Blumthaler et al., 1995; Kim et al., 2013).

3. **Total Ozone Column (TOC): How was the TOC from the Brewer instrument obtained? Additionally, why was TEMIS ozone used instead of CAMS, given that other parameters are derived from CAMS?**

**Reply**

The retrieval of total ozone from the Brewer is discussed analytically in the provided references (line 172). We do not believe that a detailed description is necessary in this paper

4. **Cloud properties: What is the spatial resolution of the COT and CMFUV products? This information is crucial for understanding the influence of shadows/3D effects and parallax errors in cloud property determination over complex terrain. Furthermore, which SEVIRI channels were used for retrieving cloud optical properties? Do they enable reliable discrimination between clouds and surface snow?**

**Reply**

Appropriate references have been provided (lines 242-243).

"A detailed description of the cloud products by MSG can be found in the relevant bibliography (Deneke et al., 2021; Météo-France, 2016)."

5. **Altitude correction for the station (line 270): How was it determined? Is it just the 5% factor per km mentioned a few lines above?**

**Reply**

Yes, it is the 5%/km correction that has been discussed earlier. We tried to make it clearer in the manuscript.

- **Results for all-sky simulations: It would be beneficial to compare CMFs in addition to UV Indices to exclude the trivial correlation arising from the diurnal SZA cycle.**

**Reply**

An extended discussion for the effect of CMF has been already performed Papachristopoulou et al., (2024). The effect of clouds is already visible in Figure 9 (Fig. 8 in the previous version), and thus we believe that adding another graph for the CMF would be beneficial. An additional discussion has been, however, added in this section to point out that the large differences shown in Fig. 9 are in all cases due to the model inability to predict accurately if the fraction of the solar disc that is occluded by clouds, especially under broken cloud conditions.

- **Classification of clear/cloudy skies: The classification, based on measured direct irradiance (lines 275–279), appears simplistic. For instance, the situation shown in the upper-right image of Fig. 5 does not seem to represent clear-sky conditions (as defined by the model). The authors could refine their classification to better account for scattered clouds and their effects on surface UV radiation.**

**Reply**

The situation that was shown in the upper right image of Figure 6 (Fig. 5 in the previous version) was not, and was not supposed to, represent clear-sky conditions. Moments before or after the time of that image there may be clear-sky conditions according to the criteria that we have set, and this is what we try to show. Figure 6 has been updated to in the context of the overall major revision of the manuscript.

We do not believe that using a more refined algorithm to distinguish occluded/unoccluded solar disc conditions would result in large differences. As discussed in the manuscript, the largest deviations seem to be due to enhancement of the UVI because of the presence of clouds (see Fig. 5) that are near the solar disc in the sky. To our knowledge there is no way to capture and consider such situations.

- **Figures: Three different types of plots are used: (i) ratios (e.g., Figs. 2–3), (ii) absolute UVI values (e.g., Figs. 6–7), and (iii) absolute differences (Figs. 8–9). Can these be more homogenised for consistency?**

**Reply**

The three different types of figures are there on purpose, since showing all three, relative differences, absolute differences, and absolute UVI values, is necessary to understand the reliability of the model, the impact of different factors that affect the UVI, and the impact of these factors when UVI is large (i.e., when it is more significant to have accurate simulations).

Nevertheless, we tried to improve the quality of the figures.

- **Other services, such as TEMIS and CAMS, already provide UV Index estimates, at least on a European scale. The authors should clarify the advantages offered by UVIOS2, such as improvements in spatial and temporal resolution.**

**Reply**

Table 2 which summarizes the main differences/similarities between the UVI from UVIOS2 and TEMIS/CAMS, and some relevant discussion have been added in Section 2.3.

- **Spectral validation: Have the authors attempted to compare the spectral output of UVIOS2 with spectra measured by QASUME-II? This could provide additional insights into the influence of various factors.**

**Reply**

We agree with the reviewer. That will be indeed very interesting and more informative regarding the impact of different factors, not only on the modeled UVI, but also on other quantities (e.g., the effective dose for the production of vitamin D). That is out of the scope of the present study, but we will do it in the future considering wider spatial and temporal scale.

**TECHNICAL REMARKS**

**- Lines 20–21: "The UVIOS2... depending on the inputs" is too general and can be removed.**

**Reply**

Done

**- Lines 76–84: The distinction between nowcasting and forecasting is unclear. If all inputs are forecasted (as suggested in lines 76–77), what differentiates nowcasting from forecasting?**

**Reply**

The reviewer is right. The word nowcasting has been deleted.

**- Line 118: "system that its basic features" → Incorrect grammar; rephrase.**

**Reply**

The manuscript has been rephrased.

**- Line 134: Remove the typo "as Heading 2."**

**Reply**

Done

**- Lines 136–139: The stated considerations apply to all radiative transfer models, not specifically to UVIOS2. The description of UVIOS2 should begin with details that are unique to this system.**

**Reply**

We believe that few introductive lines are necessary to set the discussion context for the reader before discussing the unique characteristics of UVIOS2. Thus, we did not change the manuscript at this point.

**- Figure 1: What does "% altitude" represent on the colour scale?**

**Reply**

It was a typo. It has been corrected.

**- Section 2.2 (QASUME-II vs QASUME): The uncertainty of QASUME is discussed in Sect. 2.2, but QASUME-II is used in the study. Why was QASUME not used, and why is the uncertainty of QASUME-II not discussed?**

**Reply**

What was written in the manuscript was inaccurate. We were referring to QASUME II uncertainties as it is clearly stated in revised version.

**- Table 2: A similar table for all-sky UV determination might be useful, with different headings.**

**Reply**

The only difference between the clear-sky and the all-sky simulations is the use of the UV-CMF. Thus, we do not believe that another Table for all-sky conditions would be useful. (now Table 3)

**- Figure 2: The y-axis label should clarify that the ratio represents simulated vs measured values. In addition, would plotting the ratio against solar zenith angle provide useful insights?**

**Reply**

The y-axis label has been changed. We did not find any dependence on SZA (see the following figure)

[Figure]

**- Figure 3: "ratio BG" should be replaced with "UVIOS2 / QASUME" or a similar term to clearly define the ratio. Also, why are the plots not ordered by increasing SSA?**

**Reply**

We use "BG" for the UVIOS2/QASUME ratios, with ground-based measurements as UVIOS2 inputs, as stated earlier in the manuscript. This is necessary since we have used many different input combinations. We show the graph for SSA=0.9 first, because that is the used value (i.e., this is the reference graph).

**- Lines 330–331: If the aerosol is primarily secondary, could hygroscopicity have influenced the correlation between AOD and SSA?**

**Reply**

Hygroscopicity would have probably affected the correlation between AOD and SSA in the opposite way (i.e., resulting in higher SSA values) during the transfer of the particles from Germany to Davos.

**- Line 342: Cloud-induced irradiance enhancements near the solar disk typically last for a short time. What do short-term variations in the ratio indicate about this effect?**

**Reply**

As shown in Figure 5 (added in the new version) confirms, we noticed that the increased albedo during broken cloud conditions can be very significant at Davos.

**- Lines 355–356: The treatment of limited horizon effects in the model is crucial due to the site complex orography. This information should be presented earlier. Additionally, provide an approximate (cosine-weighted) fraction of the sky obstructed by mountains.**

**Reply**

We specified in the manuscript that ~ 5% of the diffuse irradiance is blocked by the mountains. For the SZAs for which we made the comparison the direct solar beam is not blocked.

**- Figure 10: If retained, specify the UVIOS2 configuration used (in the caption).**

**Reply**

Done

**- Lines 408–409: The past tense may be more appropriate in the conclusions.**

**Reply**

Done

**- Table 3: It is unusual to introduce new results in the conclusions. Would it be better placed in the results section?**

**Reply**

Table 3 (now Table 4) and the relative discussion has been moved to Section 3.2.

**- Line 434: The term "UVIOS2 forecasts" is ambiguous here, as—if I understand correctly—clouds are not forecasted.**

**Reply**

The sentence has been deleted

[revised manuscript text omitted]

---

## Referee Report (RR1)

We thank reviewer#1 for his/her thorough and constructive review. Analytical replies to reviewer's comments are provided below. The reviewer's comments are in blue. Line numbers refer to the version with track changes.

The paper describes the UVC III campaign for calibrating and intercomparing solar UV radiometers, which was held in Davos, Switzerland, from June to August 2022, involving filter radiometers and the portable reference spectroradiometers QASUME and QASUMEII. However, the focus is on incremental improvements of the radiative transfer modeling tool (UVIOS2), which was used to forecast the UV index (UVI) with inputs from satellite, reanalysis, and ground-based sources.

Comparisons with the reference QASUME UVI measurements were used to demonstrate overall good performance of the model for clear skies, i.e., when the sun was not covered by clouds. However, much larger differences were found with instantaneous and daily UVI measurements, which were explained by cloud modeling challenges (Fig.5). Under cloud-free skies enhanced aerosol absorption, i.e., low single scattering albedo (SSA), might have explained model overestimation (Fig. 3 and 4), but there were no SSA measurements in UV to confirm this hypothesis.

**Reply**

In the revised version we used AERONET SSA (at 440 nm) data to extract safer conclusions and further support the discussion.

There is very brief mention of comparisons between QASUME and filter radiometers in section 3.3 and Figure 10 (previously published) shows that the results mainly depend on application of the consistent calibration factors (PMOD/WRC). This section needs to be either expanded or removed.

**Reply**

The UVC III campaign has been analytically described in the corresponding WMO report (Hülsen and Gröbner, 2023). This paper is mainly focused on exploiting the results of the campaign to quantify the accuracy of the UVIOS2 model. We agree that some information about the campaign should be provided for the readers' convenience, and we expanded section 3.3.

UVI references are incomplete.

**Reply**

More than 15 new references were added in the manuscript.

The paper may be suitable for publication after improving quality of the figures and completeness of the text and addressing technical questions described below.

**Reply**

We did our best to improve all aspects described above.

**RT modeling approach.**

**More details are needed describing extraterrestrial solar irradiance source, e.g., spectral smoothing applied, comparison with the state-of-the-art satellite TSIS-1 hybrid solar reference spectrum [Coddington, et al., https://doi.org/10.1029/2022EA002637 ].**

Such information has been added in the manuscript. See the corresponding reply to the specific comments below.

**The aerosols are included into the cloudless LUT (Tables 1, 2). This is different to OMI and TROPOMI satellite UVI retrievals, where aerosol and cloud effects are parameterized as a separate scattering (Cc) and absorbing (Ca) correction factors, UV = Ca(SZA, AAOD)\*Cc(SZA,COT,…)\*UVclear (SZA,TOC,…) [Arola et al., 2021 https://doi.org/10.5194/amt-14-4947-2021]. This explicit absorbing aerosol correction based on aerosol absorption optical depth (AAOD) would be especially important for North Africa and Middle East sites affected by desert dust, e.g. Roshan et al., Atmosphere 2020, 11, 96; doi:10.3390/atmos11010096.**

**Using aerosol optical thickness in UV (e.g., 340nm or 380nm) would be more appropriate as inputs to UVIOS2 model, because extrapolating visible AE would result in systematic overestimation of AOD in UV, e.g., see Fig 1 in Eck, et al., "Wavelength dependence of the optical depth of biomass burning, urban and desert dust aerosols," J. Geophys. Res. 104, 31333–31350, 1999.**

**Using cloud optical thickness in UV would be more accurate, e..g., Krotkov,et al., "Satellite estimation of spectral surface UV irradiance 2. Effects of homogeneous clouds and snow", J. Geophys. Res., http://doi.wiley.com/10.1029/2000JD900721**

**Reply**

A more accurate scheme for aerosols and clouds would increase the size of the LUT and the complexity of the simulations, and consequently the computational time which would not allow us to provide the UVI on near real time. Discussion relative to the uncertainties related to the parameterization of the spectral behavior of the absorbing aerosols, as well as with the use of the CMF instead of COT has been added in section 2.2 (lines 230 – 234, 245 - 247).

**Measurements:**

**High mountain site is not ideal for the absolute hemispherical irradiance measurements due to horizon obstruction by mountains. Provide mountain elevation**

**at the measurements site as function of the azimuth (in Figure 1) and estimate horizon blockage correction, which needs to be applied to the model and/or measurements.**

**Reply**

See our reply later on, in the corresponding specific comment.

**Clarify the difference between "clear-sky" (i.e., sun not blocked by clouds [line 275]) and "cloudless" (i.e., "clear sky", [line 180]) conditions. Provide separate comparisons statistics for completely cloud-free periods.**

**Reply**

The difference between the terms "clear-sky" and "cloudless-sky" has been clarified. A new figure (Figure 5 in the new version) has been added to further discuss the effects of clouds that do not cover the solar disc.

**Describe correction for a non-lambertian angular response of the QASUME and radiometers involved into the UVC III campaign.**

**Reply**

A detailed description of this correction has been already provided in Hulsen et al., 2016

**Technical comments:**

**Figure 1: It would be useful to add a panoramic photo of the site and angular horizon elevation table for the observation site at PMOD. Calculate the correction factor in UVIOS2 to account for the horizon blockage effect at different SZAs.**

**Reply**

We have quantified the error in the simulated UVI due to the limited horizon. Since the error is smaller than 2%, i.e., well below the overall uncertainty in our simulations, we decided not to include a correction in our model.

**Figures 2, 8-9: Add year in X-axis. Use logarithmic Y-scale. Symbols are difficult to see. Use different and larger symbols and line styles.**

**Reply**

Figures 1, 8 (now 9), and 9 (now 10) have been updated. Though, we did not use a logarithmic scale because we believe that it will make the interpretation of the results by the readers more difficult.

**55 future climatic changes – climate changes**

**Reply**

Done

**73-74. limited by the finite width of the satellite pixel – reword**

**Reply**

Done

**74 weakness of satellite sensors – need clarification**

**Reply**

We tried to clarify by adding more information after lines 73-74 (in the original version):

The accuracy of satellite-based estimates is limited due to the finite width of the satellite pixel (Kazadzis et al., 2009) and the weakness of satellite sensors to accurately probe the lower troposphere (Bais et al., 2019). In particular, assumptions are made in the satellite algorithms to describe the complex interactions between radiation, aerosols and clouds, which increase the uncertainty in the retrievals. Uncertainties in the assumed aerosol properties (Arola et al., 2021; Parisi et al., 2021), inaccurate distinction of the effect of highly reflecting terrains and cloudiness (e.g., Lakkala et al., 2020b), and uncertainties in the description of cloud cover over high-altitude sites (e.g., Schenzinger et al., 2023) are among the uncertainty sources.

**77-78: Copernicus Atmospheric Monitoring Service (CAMS) – Atmosphere**

**Reply**

Done

**100 information of the public – information to the public**

**Reply**

Done

**117 reconstructed UVI series  - reconstruct**

**Reply**

Done

**118 The UVIOS (UV-Index Operating System) nowcasting system that its basic features have been already described … - reword sentence**

**Reply**

Done

**126 summarized as follows – use colon :**

**Reply**

Done

**147 data were used as a reference**

**Reply**

Done

**168 serves as a reference**

**Reply**

Done

**197 atlas plus modtan extraterrestrial spectrum – What was spectral resolution of ETS? Was a spectral smoothing and Sun-Earth distance correction applied? Compare with the TSIS-1 HRRS [Coddington et al., https://doi.org/10.1029/2022EA002637]**

**Reply**

We thank the reviewer for pointing out the significant role of the used ETS in our results.

The spectral resolution of the used ETS is 0.05 nm, so there was no need for spectral smoothing (simulations were performed with a step of 0.5 nm). As already mentioned in the original version of the manuscript (line 317):

"The cloudless-sky UVI LUT outputs were in all cases post corrected for the effect of the varying Earth-Sun distance and for the surface elevation (1596 m for Davos)."

Regarding the use of Atlas-plus-modtran ETS, we have added the following lines in the manuscript (lines 214 - 217):

"Using a different ETS might result to differences in the simulated erythemal irradiances, as for example was shown in the study of Gröbner et al., (2017). Based on the results of the latter study we estimate that the simulated irradiances might differ by up to 5% if a different ETS was used, making the used ETS spectrum a major uncertainty factor in UVIOS2 simulations."

As suggested by the reviewer, we calculated the ratio between the erythemal irradiance that was calculated using Atlas-plus-modtran, and the erythemal irradiance that was calculated using TSIS-1 HRRS (considering wavelengths longer than 298 nm). When the latter ETS was used, erythemal irradiance at TOA was ~ 3.5% lower, which is generally in agreement to what is written in the manuscript.

**201 The US standard atmosphere (Anderson et al., 1986) was used – This model was not developed for a mountainous Davos site.**

**Reply**

Indeed, the standard US atmosphere is not necessarily representative for mountainous sites such as Davos. Nevertheless, the UVIOS2 has been developed to operate for a much wider region which includes mainly lower altitude locations. In the revised version we also added the following discussion (line 439 - 443)

"Considering invariant atmospheric properties (i.e., pressure and temperature profiles) based on a standard atmospheric profile (Anderson et al., 1986) which is not necessarily representative for a mountainous site such as Davos, introduces additional uncertainty, which however is expected to be minor relative to the overall uncertainty budget in our estimates. The used ETS and ozone absorption cross sections are more significant uncertainty factors (see Section 2.2)."

**202 the surface albedo was set to 0.05 – this may not be representative for N. Africa or Middle East sites.**

**Reply**

Indeed, there is already some discussion about surface albedo in the original version of the manuscript. The following sentence has been added (line 227):

"Adjustment of the surface albedo to the local conditions when UVIOS is used over more reflective terrains (e.g., deserts, snow-covered surfaces) is within the model improvements that are planned for the future since under such conditions assuming a standard value of 0.05 could result in large uncertainties (e.g., Weihs et al., (2001))."

**205 A correction for the effect of altitude, assuming an increase of 5% per km – There is a strong spectral dependence of the UV increase with altitude ~5% at 330nm to ~10% at 290nm, e.g., see Fig. 7 in Krotkov et al., JGR, http://doi.wiley.com/10.1029/98JD00233**

**Reply**

We agree with the reviewer. There was already discussion in the introduction about the effect of altitude. The proposed reference has been added in the introduction:

"In general, the change in the levels of the solar UV irradiance with altitude depends on atmospheric composition and has a strong wavelength dependence which is introducing difficulties in the modelling of the UVI at mountainous sites (e.g, (Dvorkin and Steinberger, 1999; Krotkov et al., 1998)). At very high-altitude (or/and latitude) sites, ice and/or snow may persist even in late spring and summer resulting in extremely high UV exposure (e.g., (Schmalwieser et al., 2017b; Siani et al., 2008; Utrillas et al., 2016))."

Nevertheless, the uncertainties due to this assumption (of the 5% increase with altitude) were quantified by performing simulations for the altitude of Davos, and for 95% of the cases the agreement was better than 2%.

**230 Analyses of different AERONET datasets shows – show**

**Reply**

Done

**231 around a typical [value]**

**Reply**

Done

**232 Given that ASY generally increases? with wavelength - ASY should decrease with wavelength**

**Reply**

It was a typo. It has been corrected

**241-244 Table 2: If input parameters are the same (SSA, ASY, surface albedo) they do not need to be included in the table.**

**Reply**

The corresponding rows have been deleted.

**245: Re-word the sentence.**

**Reply**

Done

**267 Level 2 AERONET retrievals were not used because they are not available yet. – They are available with a longer latency and could be used for reanalysis.**

**Reply**

The manuscript has been modified according to the reviewer's suggestion.

**268 nearly real time  - near real time**

**Reply**

Done

**275 For the analysis, measurements were classified as clear-sky (i.e., sun was not fully or partially covered by clouds) – This classification is not consistent with the "clear-**

sky" assumption in UVIOS2 model, where "clear-sky" is defined as "cloudless" conditions (line 182). This leads to inconsistencies in "clear sky" model to measurement comparison results.

**Reply**

The reviewer is right. The UVI has been simulated for cloudless skies (now defined as cloudless skies), while the comparison with ground based measurements has been performed for unoccluded solar disk (now defined as clear-sky). The manuscript has been revised so this is clear in the new version.

**287: Under clear-skies – This case includes scattered clouds not blocking the sun. It would be useful to show a separate comparison for the cloud-free periods in Fig 2.**

Done. Figure 5 has been added to the manuscript.

**287-288: Remove "both"**

**Reply**

Done

**Figure 2: Add Year in X-axis. Symbols are difficult to discriminate. Use different and larger symbols and different line styles. It would be useful to show cloud-free periods using different symbols.**

**Reply**

Larger symbols are used in Figures 2, 8, 9 (2,9,10 in the new version) in the revised version of the manuscript. Year has been added in X-axis. Cloud-free cases have been analyzed separately, and a new figure has been added (Figure 5 in the new version).

**Calculating average UVI ratio between DOY 190 and 200 would result in positive bias, while the bias is negative between DOY 200 and 210. Is there an explanation?**

**Reply**

The negative bias for DOY 200 – 210 is explained by the combination of the following causes:

- According to AERONET measurements much larger AOD values were recorded during DOY 200 – 210 with respect to DOY 190 – 200 (AOD at 340 nm of 0.6 or more in DOY 201-202). The SSA in these days was very high (0.98 – 0.99 at 440 nm). As can be perceived by Figure 4, the underestimation of the SSA for the modelling of the UVI was a significant factor resulting in the negative bias.
- Broken clouds (Fig. 5, 6) around the sun also resulted in enhanced real UVI, but ther role was minor relative to that of the SSA.

It is also clear from Fig. 4 that for low AOD (<0.05) conditions (as in DOY 190 - 200) the model overestimates the UVI (on average by ~ 3%) which can be due to the combined effect of uncertainties in ETS. This is possibly the reason for the (on average) positive bias during these days. Furthermore, on DOY197-199, when larger AOD values have been recorded, the SSA (AERONET level 1.5 SSA at 440 nm) was below 0.9, which again justifies part of the positive bias.

[Figure]

Figure: AOD at 500 nm from AERONET.

Part of the above discussion has been added to the revised version of the manuscript.

**295 Figure 2 shows that using highly accurate inputs for TOC, AOD at 500 nm, and AE does not result in a noticeable improvement in the accuracy of the modeled average clear-sky UVI. StDev decrease by less than 10% by using GB inputs**

**Reply**

The manuscript has been modified properly.

**302 Differences in AOD are in all cases within ± 0.1 - There are larger differences in Fig. A1**

**Reply**

Indeed, there are larger differences. The manuscript has been modified properly. The following lines have been added (line 356):

"When differences in AOD are larger (e.g., in DOY 201 – 202 the AOD from CAMS is lower by 0.15 – 0.25 relative to the AOD from CIMEL, i.e., CAMS has not captured the large AOD levels over the site) they result in correspondingly larger differences between the ratios (of 10 – 20%)."

**304-305 on average, TEMIS slightly underestimates TOC – TEMIS TOC is higher than Brewer TOC in Fig. A2**

**Reply**

Corrected

**309: differences in AOD – Use Brewer measured AOD.**

**Με σχόλια [if1]:** I am not sure what I should reply here. First of all, is the AOD from Brewer available?

**Reply**

We decided to use AERONET data together with the AE provided by AERONET for various reasons. Main reasons:

- For Calibration and uncertainty issues as AERONET uses a globally standardized sun photometer network with rigorous automatic calibration procedures, ensuring high accuracy (±0.01 to ±0.02 in AOD) while Brewer instruments rely on manual calibration. AOD below 320 nm from the Brewer, that would be more accurate for our work, is highly uncertain as discuss in the relevant bibliography.
- And mostly as AERONET spatial global coverage helps more for more "global" use of models like UVIOS2.

**313: ranging from values smaller than 0.8 (during e.g., events of dust or biomass burning aerosols – These events are not typical for Davos location. Please, provide evidence if such events did occur during UVC-III campaign.**

**Reply**

We added some discussion in this section based on the SSA values measured at 440 nm by the CIMEL. The added discussion provides additional evidence for the role of SSA. Furthermore, we added "polluted aerosols" in the parenthesis, that were possibly transferred from Germany in these days. We could not find any evidence that the site was affected by dust or smoke aerosols.

**Figure 3. Why show a hypothetical case with SSA=0.8 which is not representative for UVC-III campaign?**

**Reply**

Data from AERONET indicate that very low SSA values are possible during the campaign. Nevertheless, it is clear in our discussion that this is only a hypothesis since there are no SSA measurements in the UV available.

**327 which denotes that the SSA – which means that the SSA**

**Reply**

Done

**Figure 4. – Suggest moving this figure to supplement. You can use AERONET SSA retrievals on days 197-199.**

**Reply**

We do not agree that this figure should be moved to the supplement since it is useful for the discussion. In the revised version of the manuscript, we also used AERONET SSA in the discussion. However, we cannot draw conclusions that are solely based on the analysis of SSA values from AERONET because 1) SSA at 440 nm is not necessarily fully representative for the SSA in the region 300 – 310 nm (that mostly contributes to the UVI), and 2) because in most cases only the level 1.5 SSA product is available, that is uncertain for very low AOD values. Nevertheless, we analyzed the SSA from AEONET to strengthen our conclusions.

**354-355: Although we have not corrected the modeled UVI for the effect of limited horizon – This horizon correction should be important for Davos site. Quantify this effect using horizon elevation angle as a function of the azimuthal angle.**

**Reply**

We analyze the UVI for SZA greater than 75°, and thus the limited horizon does not affect the direct solar beam. Assuming isotropic diffuse radiation, it has been calculated that the obstacles block about 5% of the diffuse radiation. As discussed in lines 435 – 440, this loss results in an error of less than 2% for the studied cases.

**Figure 8: analysis of the outliers will be useful.**

**Reply**

The following lines have been added (lines 457 – 460): "The differences between the UVI from QASUME and the model (with both setups) are in some cases very large, reaching even values of ± 8. These large differences are mainly due to the model inability to predict accurately if the fraction of the solar disc that is occluded by clouds, especially under broken cloud conditions."

**Figure 9. The campaign average difference is close to zero, but there are certain periods (i.e., 200-210) with larger differences. Again, analysis of the largest outliers would increase the value of the comparisons.**

**Reply**

Relevant discussion has been added in the previous sections (3.1 and 3.2) which explains the larger differences. We believe that repeating the same discussion here is not necessary.

**390-400: Section 3.3 is too short. The results in Figure 10 are not discussed. Expand or remove this section.**

**Reply**

The discussion in Section 3.3 has been expanded.

**394: when the PMOD/WRC calibration – explain the difference between USER and POD/WRC calibration. Explain if radiometers were calibrated for the non-lambertian angular response (cosine correction)?**

**Reply**

Analytical discussion about these issues has been performed in Hülsen and Gröbner, 2023 and Hülsen et al., 2016.

**405 Figure 10: Text in the figure is difficult to read. Try to increase the size of the text or move the text to the caption.**

**Reply**

We increased the size of the Figure. Since Figure 10 (now Figure 11) has been provided in high resolution we believe that the text will be readable in the final version.

**426-427: when solar disc is occluded, we do not know the exact COT. – Clarify this sentence.**

**Reply**

Done

**436. shows the significance of systematic and accurate calibration of such instruments. – This is true regardless of the model performance …**

**Reply**

The sentence has been deleted

**437 discussed in previous studies – add reference to Fioletov, et al., (2004) "UV index climatology over North America from ground-based and satellite estimates", J. Geophys. Res., 109, D22308, http://doi.wiley.com/10.1029/2004JD004820**

**Reply**

Done

**440 associated to the assumptions – with the assumptions**

**Reply**

Done

**451 not available (e.g., Bais et al., 2019). – add these references:**

Krotkov, et al., "Aerosol UV absorption experiment (2002- 04): 2. Absorption optical thickness, refractive index, and single scattering albedo", Opt. Eng., 44(4), 041005, http://doi.org/10.1117/1.1886819 , 2005,

Corr, Chelsia, et al., "Retrieval of aerosol single scattering albedo at ultraviolet wavelengths at the T1 site during MILAGRO (2009)", Atmos. Chem. Phys., 9, 5813–5827, http://doi.org/10.5194/acp-9-5813-2009

Mok, J., et al., "Impacts of atmospheric brown carbon on surface UV and ozone in the Amazon Basin", Sci. Rep. (2016); https://doi.org/10.1038/srep36940

Mok, J., et al., "Comparisons of spectral aerosol absorption in Seoul, South Korea", Atmos. Meas. Tech., 11, 2295-2311, https://doi.org/10.5194/amt-11-2295-2018

Go, et al., "Ground-based retrievals of aerosol column absorption in the UV spectral region and their implications for GEMS measurements". Remote Sensing of Environment, 245, 2020, 111759,  https://doi.org/10.1016/j.rse.2020.111759

**Reply**

The recommended references have been added.

[revised manuscript text omitted]